# Revisiting Large-Scale Non-convex Distributionally Robust Optimization

**Qi Zhang[1], Yi Zhou[2], Simon Khan[3], Ashley Prater-Bennette[3], Lixin Shen[4] & Shaofeng Zou[1]**
School of Electrical, Computer and Energy Engineering, Arizona State University[1]
Department of Computer Science and Engineering, Texas A&M University[2]
Information Directorate, Air Force Research Laboratory[3]
Department of Mathematics, Syracuse University[4]
{qzhan261,zou}@asu.edu, yi.zhou@tamu.edu, lshen03@syr.edu,
{simon.khan,ashley.prater-bennette}@us.af.mil

## Abstract

Distributionally robust optimization (DRO) is a powerful technique to train robust machine learning models that perform well under distribution shifts. Compared with empirical risk minimization (ERM), DRO optimizes the expected loss under the worst-case distribution in an uncertainty set of distributions. This paper revisits the important problem of DRO with non-convex smooth loss functions. For this problem, Jin et al. (2021) showed that its dual problem is generalized $(L_0, L_1)$-smooth condition and gradient noise satisfies the affine variance condition, designed an algorithm of mini-batch normalized gradient descent with momentum, and proved its convergence and complexity. In this paper, we show that the dual problem and the gradient noise satisfy simpler yet more precise partially generalized smoothness condition and partially affine variance condition by studying the optimization variable and dual variable separately, which further yields much simpler convergence analysis. We develop a double stochastic gradient descent with clipping (D-SGD-C) algorithm that converges to an $\epsilon$-stationary point with $\mathcal{O}(\epsilon^{-4})$ gradient complexity, which matches with results in Jin et al. (2021). Our proof is much simpler, thanks to the more precise characterization of partially generalized smoothness and partially affine variance noise. We further design a variance-reduced method that achieves a lower gradient complexity of $\mathcal{O}(\epsilon^{-3})$. Our theoretical results and insights are further verified numerically on a number of tasks, and our algorithms outperform the existing DRO method (Jin et al., 2021).

## 1 Introduction

Empirical risk minimization (ERM) minimizes the expected loss under the empirical distribution $P_0$ of the training dataset with the goal of achieving a good performance on a test dataset. Though this approach yields good performance in most cases, it often times fails due to a mismatch between training and test data distributions, e.g., domain difference from training to testing in domain adaptation problems (Blitzer et al., 2006; Daume III & Marcu, 2006), imbalanced classes in the training dataset (Sagawa et al., 2019) where performance of underrepresented minority groups is important due to fairness considerations (Hashimoto et al., 2018; Grother et al., 2011); and potential adversarial attacks to the deployed model (Goodfellow et al., 2014; Sinha et al., 2017; Madry et al., 2017). For models trained using ERM, such distribution shifts will lead to significant performance degradation on test datasets.

To deal with the above challenge of potential distribution shift, distributionally robust optimization (DRO) was developed and has been widely studied in recent years (Ben-Tal et al., 2013; Shapiro, 2017; Rahimian & Mehrotra, 2019). Instead of merely optimizing the expectation of the loss function under a fixed distribution, DRO optimizes the performance over a set of probability distributions, aiming at good model performance under potential distribution shifts. Specifically, DRO assumes the test distribution lies in an uncertainty set centered in the empirical distribution $P_0$ of the

training dataset. Typically, the uncertainty set $\mathcal{U}(P_0)$ is defined as follows:

$$\mathcal{U}(P_0) = \{Q : D(Q||P_0) \leq \rho\}, \tag{1}$$

where $D$ measures the distance between $Q$ and $P_0$, e.g., Kullback-Leibler (KL) divergence, $\chi^2$ divergence or Wasserstein distance, and $\rho$ is the radius of this uncertainty set. Then the goal of DRO is to optimize the expectation of the loss function under the worst-case distribution within the uncertainty set $\mathcal{U}(P_0)$ (Rahimian & Mehrotra, 2019; Shapiro, 2017):

$$\inf_x \sup_{Q \in \mathcal{U}(P_0)} \mathbb{E}_{S \sim Q} \left[ \ell(x, S) \right], \tag{2}$$

where $x$ is the trainable model parameter, $\ell(x, S)$ is the loss function for the model with parameter $x$ and sample $S$. The formulation in equation 2 requires the optimized distribution $Q$ strictly to be inside the uncertainty set $\mathcal{U}(P_0)$, which is relatively hard to solve. In practice, it is usually preferred to use a soft penalty term, resulting in the following penalized DRO problem (Levy et al., 2020; Jin et al., 2021; Qi et al., 2021; Sinha et al., 2017):

$$\inf_x \Psi(x) := \sup_Q \mathbb{E}_{S \sim Q} \left[ \ell(x, S) \right] - \lambda D(Q||P_0). \tag{3}$$

This removes the hard constraint on $Q$ and controls the distance between the optimized distribution $Q$ and training distribution $P_0$ by a regularizer. The hyperparameter $\lambda$ is pre-selected and fixed during the training.

The DRO problems with different types of uncertainty sets, i.e., $D$'s, are fundamentally different. In this paper, we focus on a general class of distance: $\psi$-divergence distance, which includes e.g., $\chi^2$-divergence and Cressie-Read family divergence (Cressie & Read, 1984; Van Erven & Harremos, 2014). The $\psi$-divergence is widely studied in the DRO literature (Namkoong & Duchi, 2016):

$$D(Q||P_0) = \int \psi \left( \frac{dQ}{dP_0} \right) dP_0, \tag{4}$$

where $\psi$ is a non-negative convex function such that $\psi(1) = 0$ and $\psi(t) = +\infty$ for any $t < 0$.

The penalized formulation of DRO shown in equation 3 is a minimax optimization problem and is usually hard to solve. For $\psi$-divergence defined DRO problems, a popular approach is to investigate its dual formulation. By strong duality (Levy et al., 2020; Shapiro, 2017), the solution of equation 3 is equivalent to the solution of the following dual problem:

$$\inf_x \Psi(x) = \inf_{x, \eta \in \mathbb{R}} \hat{\mathcal{L}}(x, \eta) := \lambda \mathbb{E}_{S \sim P_0} \psi^* \left( \frac{\ell(x, S) - \eta}{\lambda} \right) + \eta, \tag{5}$$

where $\eta \in \mathbb{R}$ is a dual variable, $\psi^*$ is the conjugate function of $\psi$ and is defined as $\psi^*(t) = \sup_{a \in \mathbb{R}} \{ta - \psi(a)\}$. In this paper, we study this dual formulation, which has the following three **advantages** compared with the previous penalized form shown in equation 3: (i) the objective is optimized under the known training distribution $P_0$; (ii) it is easy to get an unbiased estimate of the gradient of the objective to $x$ since we do not need to take the supremum for $Q$; and (iii) it converts a minimax problem to a minimization problem, which is easier to solve.

In this paper, we focus on the DRO problem in equation 3 and equation 5 with non-convex $L$-smooth loss function $\ell$ (see Definition 2). We consider the large-scale setting, where the training dataset consists of a large number of $N$ samples. We aim to characterize the fundamental structure of this problem and develop efficient first-order algorithms with comprehensive convergence analysis.

The same problem was studied in Jin et al. (2021). It was shown that the dual objective in equation 5 is not $L$-smooth if the loss function $\ell$ is not bounded. They further show that the corresponding dual $\hat{\mathcal{L}}(x, \eta)$ satisfies the generalized smoothness condition (see Definition 3), where the Lipschitz constant grows linearly with the gradient norm $\|\nabla_{x,\eta}\hat{\mathcal{L}}\|$. Similarly, in the stochastic setting, they prove the variance of gradient estimate grows linearly with the gradient norm square $\|\nabla_{x,\eta}\hat{\mathcal{L}}\|^2$, i.e., affine variance noise (see discussion under Lemma 3). To solve this problem, a normalized momentum method was used and shown to converge with $\mathcal{O}(\epsilon^{-4})$ gradient complexity. Note that algorithms and analyses for generalized smooth optimization problems, e.g., Adagrad and Adam (Li et al., 2023; Wang et al., 2023; Zhang et al., 2024b), can also be used to solve this problem.

However, solving the problem in equation 5 as a generalized $(L_0, L_1)$-smooth optimization problem typically requires more involved convergence analysis.

In this paper, we answer the following question: **Are generalized $(L_0, L_1)$-smoothness and affine variance noise an overkill to characterize the dual $\hat{\mathcal{L}}(x, \eta)$ in equation 5?**

## 1.1 OUR CONTRIBUTIONS

Our main contributions are summarized as follows.

- We prove that the dual objective of DRO problems in equation 5 is *partially generalized smooth* (see Lemma 1), and the noise satisfies a *partially affine variance noise* condition in the stochastic setting (see Lemma 3). The above two conditions provide a much more precise and fundamental characterization of the dual in equation 5 than the generalized smoothness and affine variance noise derived in Jin et al. (2021). Such precise characterization yields much simpler convergence analysis. Our proposed more precise conditions circumvent the unbounded Lipschitz constant and unbounded noise variance challenges in the generalized $(L_0, L_1)$-smoothness problems with affine variance noises. Our conditions first show the dual problem is standard $L$-smooth in the dual variable $\eta$ and the noise of gradient on $\eta$ has bounded variance. Then under our partially generalized smoothness and partially affine variance noise condition, the Lipschitz constant and noise variance on the model parameter $x$ are also bounded, making the objective easy to solve.

- We show that in the deterministic setting, an algorithm as simple as gradient descent can solve the problem with iteration complexity of $\mathcal{O}(\epsilon^{-2})$; and in the stochastic setting, an algorithm as simple as double stochastic gradient descent with clipping (D-SGD-C) can solve the problem with gradient complexity of $\mathcal{O}(\epsilon^{-4})$. We further design a double spider with clipping (D-Spider-C) algorithm and show its convergence with an improved $\mathcal{O}(\epsilon^{-3})$ gradient complexity. Thanks to our more precise characterizations of *partially generalized smooth* and *partially affine variance noise*, our analyses are much simpler than those in Jin et al. (2021) and those for Spider algorithms (Chen et al., 2023; Reisizadeh et al., 2023), which are developed merely for general generalized smooth problems with affine variance noise and are not tailored specially for DRO problems.

- Our methods converge with computational complexities independent of the number of training samples $N$, and thus are applicable to large-scale training datasets.

- Numerical results are conducted to verify our theoretical results. We observe that our proposed algorithms outperform the existing DRO method (Jin et al., 2021).

## 1.2 RELATED WORKS

**DRO.** *Scalability:* Many existing approaches are not scalable when the training dataset is large. The method (Namkoong & Duchi, 2016) is not feasible for large-scale problems, which parameterizes the unknown distribution by a vector of dimension $N$ and models the DRO problem as a minimax optimization problem. Following this method, many minimax methods such as Rafique et al. (2022); Lin et al. (2020); Xu et al. (2023) can be used to address the DRO problem. However, a computational complexity that is linear (or even worse) in the size of the training set is not preferable for large-scale problems. In this paper, our stochastic algorithms have a per-iteration complexity that is independent of the training dataset.

*Convex loss functions:* Some existing methods (Duchi & Namkoong, 2018; Levy et al., 2020; Wang et al., 2021; 2024; Hashimoto et al., 2018) require the loss function to be convex, which, however, fail to capture a wide range of machine learning problems where the loss function is non-convex, e.g., neural networks. In this paper, we focus on the general non-convex smooth loss function.

*Bounded loss functions:* For the non-convex DRO problem, existing studies, e.g., Qi et al. (2021; 2022); Zhang et al. (2024a); Soma & Yoshida (2020) require the loss function $\ell$ to be bounded (or even more restricted assumptions). In this paper, we focus on non-convex smooth loss functions, which may potentially be unbounded.

*Non-convex smooth loss functions:* Jin et al. (2021) is the first study for non-convex DRO problems with general $\psi$-divergence defined uncertainty sets in large-scale settings. By combining all trainable

parameters together $z = (x, \lambda)$, the authors prove that the dual objective shown in equation 5 is generalized $(L_0, L_1)$-smooth in $z$. Then a normalized-momentum method is designed and an $\epsilon$-stationary point is guaranteed with a computational complexity of $\mathcal{O}(\epsilon^{-4})$. In this paper, we share the same setting with Jin et al. (2021) and we provide a simpler yet more precise characterization of the dual problem: partially generalized smoothness and partially affine variance noise. We show that a simple SGD type algorithm finds an $\epsilon$-stationary point with a complexity of $\mathcal{O}(\epsilon^{-4})$. Moreover, we design a Spider algorithm, reducing the complexity to $\mathcal{O}(\epsilon^{-3})$.

**Generalized $(L_0, L_1)$-smoothness.** As Jin et al. (2021) pointed out, the non-convex DRO problem can be solved as a generalized $(L_0, L_1)$-smooth optimization problem, which, however, introduces unnecessary complications to theoretical analysis for the DRO problem in this paper. Nevertheless, we briefly review algorithms and analyses for generalized smooth optimization problems in the literature below. The generalized $(L_0, L_1)$-smoothness problem is first introduced in Zhang et al. (2019), where a clipping method is investigated. However, for the stochastic setting, the gradient estimation error is required to be bounded almost surely. In this paper, our method works for noise with partially affine variance (shown in Lemma 3), which is a much weaker condition and more precise characterization for this DRO problem. Modern methods such as normalized-momentum (Jin et al., 2021), Adagrad (Wang et al., 2023), and Adam (Li et al., 2023; Zhang et al., 2024b) are studied for generalized $(L_0, L_1)$-smoothness problem, and they can also be used to solve the DRO problem in this paper. In our paper, we show that for the DRO problem, simple SGD can get the same stationary point with the same gradient complexity. To reduce the complexity, Spider (Fang et al., 2018) is studied for the generalized $(L_0, L_1)$-smoothness problem (Chen et al., 2023; Reisizadeh et al., 2023). In this paper, based on our precise characterizations of partially generalized smoothness and partially affine variance noise, our proof is much simpler than Chen et al. (2023). Moreover, we show the gradient converges in expectation, which is stronger than the convergence with high probability in Reisizadeh et al. (2023).

## 2 PRELIMINARIES

Denote by $s$ a sample in $\mathbb{S}$ and let $P_0$ be the empirical distribution of the $N$ training samples $\{s_i\}_{i=1}^N$. In the large-scale setting studied in this paper, we assume the number of training samples $N$ is extremely large. We use $\|\cdot\|$ to denote the Euclidean norm and $\langle\cdot,\cdot\rangle$ to denote the standard dot product. Define a function $(a)_+ = \max(a, 0)$. For a set $C$, $\mathbb{1}_C$ is an indicator function such that $\mathbb{1}_C(a) = 0$ if $a \in C$ and $\mathbb{1}_C(a) = +\infty$ otherwise. Let $x \in \mathbb{R}^d$ be the trainable parameters where $d$ is the dimension. The loss function is defined as $\ell : \mathbb{R}^d \times \mathbb{S} \to \mathbb{R}$. For a differentiable function $f : \mathbb{R}^d \to \mathbb{R}$, $x$ is an $\epsilon$-stationary point if $\|\nabla_x f(x)\| \leq \epsilon$, where $\nabla_x f(x)$ is the gradient of $f$ to $x$. For a random vector $X$, denote by $\mathbb{E}$ the expectation and $\mathbb{V}$ the sum of its element-wise variance, where $\mathbb{V}(X) := \mathbb{E}[\|X - \mathbb{E}[X]\|^2]$. We further provide some definitions.

**Definition 1** (Lipschitz continuous)**.** *A function $f : \mathbb{R}^d \to \mathbb{R}$ is called $G$-Lipschitz continuous if for any $x, y \in \mathbb{R}^d$, $|f(x) - f(y)| \leq G\|x - y\|$, where $G > 0$ is some finite constant.*

**Definition 2** (Standard $L$-smooth)**.** *A differentiable function $f : \mathbb{R}^d \to \mathbb{R}$ is $L$-smooth if for any $x, y \in \mathbb{R}^d$, $\|\nabla_x f(x) - \nabla_x f(y)\| \leq L\|x - y\|$, where $L > 0$ is some finite constant.*

These two definitions cover a wide range of problems in optimization studies. Recently, a generalized $(L_0, L_1)$-smoothness condition is proposed (Zhang et al., 2019; Chen et al., 2023; Li et al., 2023), which is strictly weaker than the standard $L$-smoothness condition.

**Definition 3** (Generalized $(L_0, L_1)$-smooth)**.** *A differentiable function $f : \mathbb{R}^d \to \mathbb{R}$ is generalized $(L_0, L_1)$-smooth if for any $x, y \in \mathbb{R}^d$, we have that $\|\nabla_x f(x) - \nabla_x f(y)\| \leq (L_0 + L_1\|\nabla_x f(x)\|)\|x - y\|$, where $L_0, L_1 > 0$ are some finite constants.*

Note that there are two version of the $(L_0, L_1)$-smoothness, one requires that the inequality only applies to $\|x - y\| \leq \frac{1}{L_0}$ (Zhang et al., 2019) and one does not require that (Chen et al., 2023). In DRO setting, it can be proved that for the dual objective Jin et al. (2021), the inequality holds for any $x, y \in \mathbb{R}^d$ thus in this paper we follow the second definition.

In this paper, we focus on a non-convex and smooth loss function $\ell$.

**Assumption 1.** *For any sample $s \in \mathbb{S}$, the loss function $\ell(x, s)$ is $G$-Lipschitz continuous and $L$-smooth in $x$.*

We further make the following assumption on the $\psi$-divergence.

**Assumption 2.** *The conjugate function $\psi^*$ of $\psi$ is $M$-smooth.*

Assumption 2 can be satisfied by a wide range of $\psi$-divergences (see Table 1).

| Divergence | $\psi(t)$ | $\psi^*(t)$ |
|---|---|---|
| $\chi^2$ | $\frac{1}{2}(t-1)^2$ | $-1 + \frac{1}{4}(t+2)_+^2$ |
| KL-regularized CVaR | $\mathbb{1}_{[0,\alpha^{-1})} + t\log(t) - t + 1, \alpha \in (0,1)$ | $\min(e^t, \alpha^{-1}(1 + t + \log(\alpha))) - 1$ |
| Cressie-Read | $\frac{t^k - tk + k - 1}{k(k-1)}, k \in \mathbb{R}$ | $\frac{1}{k}\left(((k-1)t+1)_+^{\frac{k}{k-1}} - 1\right)$ |

Table 1: Commonly used divergences with $L$-smooth conjugates.

The goal of this paper is to find an $\epsilon$-stationary point of the penalized DRO problem in equation 3, which is a minimax optimization problem and is usually hard to solve. For $\psi$-divergence defined DRO problems, a popular approach is to investigate its dual formulation. By strong duality (Levy et al., 2020; Shapiro, 2017), we have that

$$\Psi(x) = \inf_{\eta \in \mathbb{R}} \hat{\mathcal{L}}(x, \eta) := \lambda \mathbb{E}_{S \sim P_0} \psi^*\left(\frac{\ell(x, S) - \eta}{\lambda}\right) + \eta. \tag{6}$$

Under Assumptions 1 and 2, it can be shown that $\Psi(x)$ is differentiable (Jin et al., 2021). Define $\mathcal{L}(x, \eta) = \hat{\mathcal{L}}(x, G\eta)$. Then $\|\nabla_{x,\eta}\mathcal{L}(x, \eta)\| \le \epsilon/\sqrt{2}$ implies that $\|\nabla\Psi(x)\| \le \epsilon$ (Jin et al., 2021). Thus, it is equivalent to find an $\epsilon$-stationary point of $\mathcal{L}(x, \eta)$.

# 3 MAIN RESULTS

## 3.1 PARTIALLY GENERALIZED SMOOTHNESS

Let $z = (x, \eta)$. Approach in Jin et al. (2021) directly optimizes over $z$, where it was shown that $\mathcal{L}(z)$ is generalized $(L_0, L_1)$-smooth in $z$ with $L_0 = L + 2G^2\lambda^{-1}M$ and $L_1 = L/G$. In the following, we provide a more precise characterization of $\mathcal{L}$ by separately studying $x$ and $\eta$.

**Lemma 1** (Partially generalized $(L_0, L_1, L_2)$-smoothness). *Under Assumptions 1 and 2, $\mathcal{L}(x, \eta)$ is $(L_0, L_1)$-partially smooth in $x$ and $L_2$-smooth in $\eta$ such that for any $x, x' \in \mathbb{R}^d$ and $\eta, \eta' \in \mathbb{R}$ we have that*

$$\|\nabla_x\mathcal{L}(x, \eta) - \nabla_x\mathcal{L}(x', \eta)\| \le \underbrace{(L_0 + L_1|\nabla_\eta\mathcal{L}(x, \eta)|)}_{term\ (a)} \|x - x'\|, \tag{7}$$

$$|\nabla_\eta\mathcal{L}(x, \eta) - \nabla_\eta\mathcal{L}(x, \eta')| \le L_2|\eta - \eta'|, \tag{8}$$

*where $L_0 = G + \frac{G^2 M}{\lambda}$, $L_1 = \frac{L}{G}$ and $L_2 = \frac{G^2 M}{\lambda}$.*

The proof is available in Appendix A.1. Observe that $\mathcal{L}(x, \eta)$ is smooth in $\eta$ for any $x$. Thus, optimizing over $\eta$ should not be as hard as solving a generalized smooth problem. Moreover, in equation 7, the Lipschitz constant in $x$ (term (a)) is linear in the gradient to $\eta$: $\nabla_\eta\mathcal{L}(x, \eta)$, but does not depend on the gradient to $x$: $\nabla_x\mathcal{L}(x, \eta)$. Compared with the generalized $(L_0, L_1)$-smoothness used in Jin et al. (2021), Lemma 1 provides a more precise characterization of $\mathcal{L}$. Intuitively, due to the smoothness in $\eta$, one can expect a quick find of a point with a bounded gradient to $\eta$. Consequently, the Lipschitz constant in $x$ will also become bounded, which circumvents the unbounded Lipschitz constant challenges in generalized $(L_0, L_1)$-smoothness problems and makes the objective easier to optimize.

**Remark 1.** *This partially generalized $(L_0, L_1, L_2)$-smoothness condition is weaker than the standard $L$-smoothness condition but stronger than the generalized $(L_0, L_1)$-smoothness condition.*

## 3.2 DETERMINISTIC SETTING

To warm up, we first consider the deterministic setting. We first propose a double gradient descent (D-GD) algorithm, which updates $x$ and $\eta$ alternatively (see Algorithm 1). This is in contrast to

the approach in Jin et al. (2021) where $x$ and $\eta$ are optimized jointly. The key idea is to leverage the standard $L$-smoothness property in $\eta$ to bound $\nabla_\eta \mathcal{L}(x, \eta)$, to reduce equation 7 to a smooth condition, and then to bound $\nabla_x \mathcal{L}(x, \eta)$.

---

**Algorithm 1** D-GD

---

**Input:** initialization $x_0, \eta_0$, step sizes $\alpha_t, \beta_t$, number of iterations $T$

1: $t \leftarrow 0$
2: **while** $t \leq T - 1$ **do**
3:    $\eta_{t+1} \leftarrow \eta_t - \alpha_t \nabla_\eta \mathcal{L}(x_t, \eta_t)$
4:    $x_{t+1} \leftarrow x_t - \beta_t \nabla_x \mathcal{L}(x_t, \eta_{t+1})$
5:    $t \leftarrow t + 1$
6: **end while**

---

### 3.2.1 DOUBLE GRADIENT DESCENT (D-GD)

In this section, we choose constant step sizes $\alpha_t, \beta_t$ in Algorithm 1. In the following theorem, we show that D-GD converges to an $\epsilon$-stationary point with an iteration complexity of $\mathcal{O}(\epsilon^{-2})$.

**Theorem 1.** *Let $H = 2L_2(\mathcal{L}(x_0, \eta_0) - \inf_{x, \eta} \mathcal{L}(x, \eta))$ and $L' = L_0 + L_1 \sqrt{H}$. Set $\alpha_t = \frac{1}{L_2}, \beta_t = \frac{1}{L'}, T \geq 8 \max(L_2, L') \frac{\mathcal{L}(x_0, \eta_0) - \inf_{x, \eta} \mathcal{L}(x, \eta)}{\epsilon^2}$. For Algorithm 1, we then have that*

$$\min_{t < T} \|\nabla_{x, \eta} \mathcal{L}(x_t, \eta_{t+1})\| \leq \sqrt{2}\epsilon.$$

The proof is available in Appendix A.2. One key step in the proof is the following descent lemma:

**Lemma 2** (Descent lemma). *For the partially generalized $(L_0, L_1, L_2)$-smooth function $\mathcal{L}(x, \eta)$ defined in Lemma 1, we have that for any $x, x' \in \mathbb{R}^d$ and $\eta \in \mathbb{R}$,*

$$\mathcal{L}(x', \eta) \leq \mathcal{L}(x, \eta) + \langle \nabla_x \mathcal{L}(x, \eta), x' - x \rangle + \frac{L_0 + L_1 |\nabla_\eta \mathcal{L}(x, \eta)|}{2} \|x - x'\|^2. \tag{9}$$

The proof can be found in Appendix A.3. In the proof, by setting $\alpha_t = \frac{1}{L_2}$ and using the standard $L_2$-smoothness in $\eta$, at each step $t$ we can show that $\mathcal{L}(x_t, \eta_{t+1}) \leq \mathcal{L}(x_t, \eta_t)$. Moreover, since $\mathcal{L}$ is $L_2$-smooth in $\eta$, we have that for any $x, \eta$,

$$|\nabla_\eta \mathcal{L}(x, \eta)|^2 \leq 2L_2 \left( \mathcal{L}(x, \eta) - \inf_{\eta'} \mathcal{L}(x, \eta') \right) \leq 2L_2 \left( \mathcal{L}(x, \eta) - \inf_{x', \eta'} \mathcal{L}(x', \eta') \right), \tag{10}$$

which indicates that a bounded function value implies a bounded gradient of $\eta$.

We then prove this by mathematical induction. It can be easily shown using equation 10 that at $t = 0$, $|\nabla_\eta \mathcal{L}(x_t, \eta_t)| \leq \sqrt{H}$ and $|\nabla_\eta \mathcal{L}(x_t, \eta_{t+1})| \leq \sqrt{H}$. By Lemma 1, it can be shown that the Lipschitz constant to $x$ at $(x_t, \eta_{t+1})$ is upper bounded by $L' = L_0 + L_1 \sqrt{H}$. By setting $\beta_t = \frac{1}{L'}$, we can show $\mathcal{L}(x_{t+1}, \eta_{t+1}) \leq \mathcal{L}(x_t, \eta_{t+1})$. Thus at $t + 1$ we also have that $|\nabla_\eta \mathcal{L}(x_{t+1}, \eta_{t+1})| \leq \sqrt{H}$ and $|\nabla_\eta \mathcal{L}(x_{t+1}, \eta_{t+2})| \leq \sqrt{H}$. This further implies that the Lipschitz constant to $x$ at $(x_{t+1}, \eta_{t+2})$ is upper bounded by $L'$. Therefore, by induction, along the training trajectory, the objective function $\mathcal{L}(x, \eta)$ is $L_2$-smooth in $\eta$ and $L'$-smooth in $x$. We then convert the DRO problem to a standard $L$-smooth optimization problem, which is much easier to address.

### 3.2.2 DOUBLE GRADIENT DESCENT WITH CLIPPING (D-GD-C)

Algorithm 1 with fixed stepsizes is straightforward and convenient to be employed in practice. However, Theorem 1 indicates that the computational complexity is linear in $L'$. In this section, we study Algorithm 1 with an adaptive step size for the update of $x$, which we refer to as Double Gradient Descent with Clipping (D-GD-C). In the following theorem, we provide the convergence guarantee:

**Theorem 2.** *Set $\alpha_t = \frac{1}{L_2}, \beta_t = \min\left( \frac{1}{2L_0}, \frac{1}{2L_1 |\nabla_\eta \mathcal{L}(x_t, \eta_{t+1})|} \right), T \geq \frac{\mathcal{L}(x_0, \eta_0) - \inf_{x, \eta} \mathcal{L}(x, \eta)}{\epsilon^2} \max(8L_2, 16L_0)$ for Algorithm 1. For $\epsilon \leq \frac{L_0}{L_1}$, we then have that*

$$\min_{t < T} \|\nabla_{x, \eta} \mathcal{L}(x_t, \eta_{t+1})\| \leq \sqrt{2}\epsilon.$$

The detailed proof is available in Appendix A.4. Due to the adaptive design of $\beta_t$, the step size is small when the gradient to $\eta$ is large. Thus the function value is decreasing and we can prove that for some $t < T$, $\alpha_t(\nabla_\eta \mathcal{L}(x_t, \eta_t))^2 + \beta_t \|\nabla_x \mathcal{L}(x_t, \eta_{t+1})\|^2 \leq \mathcal{O}(\epsilon^2)$.

If using the analysis for generalized $(L_0, L_1)$-smooth optimization, e.g., in Jin et al. (2021), it can be shown that $\alpha'_t \|\nabla_z \mathcal{L}(z)\|^2 \leq \mathcal{O}(\epsilon^2)$, where $\alpha'_t$ is an adaptive step size and is a function of $\|\nabla_z \mathcal{L}(z)\|$. Thus, $\alpha'_t \|\nabla_z \mathcal{L}(z)\|^2$ is a function of $\|\nabla_z \mathcal{L}(z)\|$ and it will need non-trivial efforts to show that $\|\nabla_z \mathcal{L}(z)\|$ is bounded. In contrast, using our more precise partial $(L_0, L_1, L_2)$-smoothness characterization, since $\alpha_t$ is a constant and $\mathcal{L}$ is $L_2$-smooth in $\eta$, it can be shown that $|\nabla_\eta \mathcal{L}(x_t, \eta_{t+1})|$ is bounded. We then obtain a lower bound on $\beta_t$: from $|\nabla_\eta \mathcal{L}(x_t, \eta_{t+1})| \leq \epsilon \frac{L_0}{L_1}$, it follows that $\beta_t = \frac{1}{2L_0}$ which is a constant. Thus, one can easily prove $\|\nabla_x \mathcal{L}(x_t, \eta_{t+1})\|$ is bounded if $\beta_t \|\nabla_x \mathcal{L}(x_t, \eta_{t+1})\|^2$ is bounded. This implies an $\epsilon$-stationary point.

## 3.3 STOCHASTIC SETTING

In the deterministic setting, the per-iteration gradient complexity is $N$, and therefore, the approach may not be scalable when $N$ is large. In this section, we develop stochastic gradient algorithms that have per-iteration gradient complexity independent of the training dataset size $N$. For the loss function, we follow the most relaxed bounded variance assumption in the $\psi$-divergence DRO literature (Jin et al., 2021):

**Assumption 3** (Bounded variance). *For any $x \in \mathbb{R}^d$ and $S \sim P_0$, the variance of the loss function is bounded:*

$$\mathbb{E}_{S \sim P_0}[(\ell(x, S) - \ell(x))^2] \leq \sigma^2, \tag{11}$$

*where $\sigma$ is a constant and $\ell(x) = \mathbb{E}_{S \sim P_0}[\ell(x, S)]$.*

We note that stronger assumptions of bounded loss function, i.e., $\ell(x, s)$ is bounded for any $x, s$ are used in $\psi$-divergence DRO literature (Qi et al., 2022; Levy et al., 2020; Zhang et al., 2024a). However, their approaches cannot be easily extended to unbounded loss functions.

We then provide the following lemma to characterize the variance of the stochastic gradient of $\mathcal{L}(x, \eta)$ to $x$ and $\eta$. Define $\mathcal{L}(x, \eta, S) = \lambda \psi^* \left( \frac{\ell(x, S) - G\eta}{\lambda} \right) + G\eta$.

**Lemma 3** (Partially affine variance noise). *Under Assumptions 1, 2 and 3, for any $x \in \mathbb{R}^d$ and $\eta \in \mathbb{R}$, we have that*

$$\mathbb{V}_{S \sim P_0}[\nabla_x \mathcal{L}(x, \eta, S)] \leq D_0 + D_1 \left( \nabla_\eta \mathcal{L}(x, \eta) \right)^2, \tag{12}$$

$$\mathbb{V}_{S \sim P_0}[\nabla_\eta \mathcal{L}(x, \eta, S)] \leq D_2 \tag{13}$$

*where $D_0 = 8G^2 + 10G^2 M^2 \lambda^{-2} \sigma^2$, $D_1 = 8$ and $D_2 = G^2 M^2 \lambda^{-2} \sigma^2$.*

The proof is available in Appendix A.5. Note that Lemma 3 provides a much more precise characterization of the stochastic gradient variance than the affine variance condition in equation 14 (Jin et al., 2021) :

$$\mathbb{V}_{S \sim P_0}[\nabla_z \mathcal{L}(z, S)] \leq D'_0 + D'_1 \|\nabla_z \mathcal{L}(z)\|^2, \tag{14}$$

where $D'_0, D'_1$ are some positive constants. The affine variance condition in Jin et al. (2021) is challenging to analyze since $\mathcal{L}$ is $(L_0, L_1)$-smooth in $z$ and $\|\nabla_z \mathcal{L}(z)\|$ is non-trivial to bound. In contrast, our Lemma 3 indicates that the stochastic gradient to $\eta$ has a bounded variance, and the variance of the stochastic gradient to $x$ is linear only in $(\nabla_\eta \mathcal{L})^2$. Since the dual objective is standard $L_2$-smooth in $\eta$ and the gradient noise has a bounded variance, we can easily bound $(\nabla_\eta \mathcal{L})^2$ and show the variance of the stochastic gradient to $\eta$ is also bounded.

### 3.3.1 DOUBLE STOCHASTIC GRADIENT DESCENT WITH CLIPPING (D-SGD-C)

In this section, we develop a D-SGD-C algorithm that updates $x$ and $\eta$ alternatively (Algorithm 2). In the stochastic setting, we use a mini-batch of samples to estimate the gradient, and $N_1, N_2$ are the batch sizes. For a mini-batch of samples $\mathcal{B} = \{\xi^{(i)}\}_{i=1}^{|\mathcal{B}|}$, denote by

$$\mathcal{L}(x, \eta, \mathcal{B}) = \sum_{i=1}^{|\mathcal{B}|} \frac{1}{|\mathcal{B}|} \left( \lambda \psi^* \left( \frac{\ell\left(x, \xi^{(i)}\right) - G\eta}{\lambda} \right) + G\eta \right)$$

an estimate of $\mathcal{L}(x, \eta)$, where $|\mathcal{B}|$ is the batch size.

---

**Algorithm 2** D-SGD-C

---

**Input:** initialization $(x_0, \eta_0)$, step sizes $\alpha_t, \beta_t$, number of interactions $T$, batch sizes $N_1, N_2$
 1: $t \leftarrow 0$
 2: **while** $t \leq T - 1$ **do**
 3:     Draw $N_1$ i.i.d. samples $\mathcal{B}_1$ from $P_0$ and compute $g_t \leftarrow \nabla_\eta \mathcal{L}(x_t, \eta_t, \mathcal{B}_1)$
 4:     $\eta_{t+1} \leftarrow \eta_t - \alpha_t g_t$
 5:     Draw $N_2$ i.i.d. samples $\mathcal{B}_2$ from $P_0$ and compute $v_t \leftarrow \nabla_x \mathcal{L}(x_t, \eta_{t+1}, \mathcal{B}_2)$
 6:     $x_{t+1} \leftarrow x_t - \beta_t v_t$
 7:     $t \leftarrow t + 1$
 8: **end while**

---

The following theorem establishes the convergence and complexity of Algorithm 2.

**Theorem 3.** *Set $\alpha = \frac{1}{2L_2}, \beta_t = \min(\frac{1}{2L_0}, \frac{\epsilon}{L_0\|v_t\|})$, $T, N_1, N_2 = \mathcal{O}(\epsilon^{-2})$ for Algorithm 2. We then have that*

$$\min_{t < T} \mathbb{E}[\|\nabla_{x,\eta} \mathcal{L}(x_t, \eta_{t+1})\|] \leq 6\epsilon.$$

Exact expressions of $T, N_1, N_2$ and the proof are available in Appendix A.6. In Algorithm 2, we choose adaptive $\beta_t \leq \frac{\epsilon}{L_0\|v_t\|}$ in order to bound the additional term of $|\nabla_\eta \mathcal{L}(x_t, \eta_{t+1})|\|x_t - x_{t+1}\|^2$ term in Lemma 1 introduced by partially generalized smoothness. The rest terms in the decent lemma are the same as standard $L$-smooth problems; thus, we can find some $t < T$ such that $\mathbb{E}[\beta_t\|v_t\|^2] \leq \mathcal{O}(\epsilon^2)$. By the definition of $\beta_t$, it can be shown that $\beta_t\|v_t\|^2 \geq \frac{\epsilon\|v_t\|}{L_0} - \frac{\epsilon^2}{2L_0}$ for any $t$. Thus we can find some $t < T$ such that $\mathbb{E}[\|v_t\|] \leq \mathcal{O}(\epsilon)$.

Using this fact, we now show that $\mathbb{E}[\|v_t - \nabla_x \mathcal{L}(x_t, \eta_{t+1})\|] \leq \mathcal{O}(\epsilon)$. Based on Lemma 3 and $N_2 = \mathcal{O}(\epsilon^{-2})$, we have that $\mathbb{V}[v_t] \leq \mathcal{O}(\epsilon^2 + \epsilon^2(\nabla_\eta \mathcal{L}(x_t, \eta_{t+1}))^2)$. Moreover, $\mathbb{E}[\|v_t - \nabla_x \mathcal{L}(x_t, \eta_{t+1})\|] \leq \sqrt{\mathbb{E}[\|v_t - \nabla_x \mathcal{L}(x_t, \eta_{t+1})\|^2]} = \sqrt{\mathbb{V}[v_t]}$, implying that $\mathbb{E}[\|v_t - \nabla_x \mathcal{L}(x_t, \eta_{t+1})\|] \leq \mathcal{O}(\epsilon + \epsilon|\nabla_\eta \mathcal{L}(x_t, \eta_{t+1})|)$. In Lemma 1, we show the dual objective is $L_2$-smooth in $\eta$; thus, it is less challenging to bound $|\nabla_\eta \mathcal{L}(x_t, \eta_{t+1})|$. We then can find some $t < T$ such that $\mathbb{E}[\|\nabla_x \mathcal{L}(x_t, \eta_{t+1})\|] \leq \mathbb{E}[\|v_t - \nabla_x \mathcal{L}(x_t, \eta_{t+1})\|] + \mathbb{E}[\|v_t\|] \leq \mathcal{O}(\epsilon)$ and can complete the proof. We also provide Algorithm 4, which uses momentum method and does not require mini-batches (See Appendix A.11).

### 3.3.2 DOUBLE SPIDER WITH CLIPPING (D-SPIDER-C)

Theorem 3 indicates a computational complexity of $\mathcal{O}(\epsilon^{-4})$ is required by D-SGD-C to find an $\epsilon$-stationary point of $\Psi(x)$. This complexity can be further improved by our following variance-reduced method (Algorithm 3), which we call Double Spider with Clipping (D-Spider-C). In this method, we first compute our estimate of the gradient to $\eta$ $(x)$ with a large batch of samples with size of $N_1$ $(N_3)$. We then update our estimate with a small batch of samples with a size of $N_2$ $(N_4)$. For every $q$ iteration, we refresh our estimate with a large batch of samples with size of $N_1$ $(N_3)$.

To analyze Algorithm 3, we further develop the following property of gradients $\nabla_x \mathcal{L}(x, \eta, s)$ and $\nabla_\eta \mathcal{L}(x, \eta, s)$.

**Lemma 4.** *For any $x, x' \in \mathbb{R}^d, \eta, \eta' \in \mathbb{R}$ and $s \in \mathbb{S}$, $\nabla_\eta \mathcal{L}(x, \eta, s)$ is $L_2$-continuous in x:*

$$|\nabla_\eta \mathcal{L}(x, \eta, s) - \nabla_\eta \mathcal{L}(x', \eta, s)| \leq L_2\|x - x'\| \qquad (15)$$

*and $\nabla_x \mathcal{L}(x, \eta, s)$ is $L_2$-continuous in $\eta$:*

$$\|\nabla_x \mathcal{L}(x, \eta, s) - \nabla_x \mathcal{L}(x, \eta', s)\| \leq L_2|\eta - \eta'|. \qquad (16)$$

The proof is available in Appendix A.7. Lemma 4 theoretically characterize how the update on $x$ (or $\eta$) changes the gradient to $\eta$ (or $x$). We then provide the theorectical result.

---

**Algorithm 3** D-Spider-C

---

**Input:** initialization $(x_0, \eta_0)$, step sizes $\alpha_t, \beta_t$, epoch size $q$, number of iterations $T$, batch sizes $N_1, N_2, N_3, N_4$

 1: **while** $t \leq T - 1$ **do**
 2:     **if** $t \mod q == 0$ **then**
 3:         Draw $N_1$ i.i.d. samples $\mathcal{B}_1$ from $P_0$ and compute $g_t \leftarrow \nabla_\eta \mathcal{L}(x_t, \eta_t, \mathcal{B}_1)$
 4:     **else**
 5:         Draw $N_2$ i.i.d. samples $\mathcal{B}_2$ from $P_0$ and compute $g_t \leftarrow \nabla_\eta \mathcal{L}(x_t, \eta_t, \mathcal{B}_2) - \nabla_\eta \mathcal{L}(x_{t-1}, \eta_{t-1}, \mathcal{B}_2) + g_{t-1}$
 6:     **end if**
 7:     $\eta_{t+1} \leftarrow \eta_t - \alpha_t g_t$
 8:     **if** $t \mod q == 0$ **then**
 9:         Draw $N_3$ i.i.d. samples $\mathcal{B}_3$ from $P_0$ and compute $v_t \leftarrow \nabla_x \mathcal{L}(x_t, \eta_{t+1}, \mathcal{B}_3)$
10:     **else**
11:         Draw $N_4$ i.i.d. samples $\mathcal{B}_4$ from $P_0$ and compute $v_t \leftarrow \nabla_x \mathcal{L}(x_t, \eta_{t+1}, \mathcal{B}_4) - \nabla_x \mathcal{L}(x_{t-1}, \eta_t, \mathcal{B}_4) + v_{t-1}$
12:     **end if**
13:     $x_{t+1} \leftarrow x_t - \beta_t v_t$
14:     $t \leftarrow t + 1$
15: **end while**

---

**Theorem 4.** *Set* $\alpha = \frac{1}{4L_2}, \beta_t = \min(\frac{1}{2L_0}, \frac{\epsilon}{L_0\|v_t\|}), T, N_1, N_3 = \mathcal{O}(\epsilon^{-2})$ *and* $N_2, N_4 = \mathcal{O}(q)$ *for Algorithm 3. We have that for some constant* $c_4 > 0$

$$\min_{t<T} \mathbb{E}[\|\nabla_{x,\eta}\mathcal{L}(x_t, \eta_{t+1})\|] \leq c_4\epsilon.$$

The full version and its proof are available in Appendix A.8. From the result we can show that the total gradient complexity is $\mathcal{O}\left(\frac{\epsilon^{-4}}{q} + \epsilon^{-2}q\right)$. By choosing $q = \mathcal{O}(\epsilon^{-1})$, we can reduce the total gradient complexity from $\mathcal{O}(\epsilon^{-4})$ to $\mathcal{O}(\epsilon^{-3})$. Compared with Algorithm 2, the biggest difference is how we design our estimators $g_t$ and $v_t$. Rather than using unbiased gradient estimators, Algorithm 3 introduces momentum in the updates of $g_t$ and $v_t$, which, however, leads to biased gradient estimates. The key challenge in our proof is to bound such bias (see the following lemma).

**Lemma 5.** *Using the same parameters as in Theorem 4, for any* $t_0 < T$ *such that* $t_0 \mod q = 0$*, we have that*

$$\sum_{t=t_0}^{t_0+q-1} \mathbb{E}\left[(g_t - \nabla_\eta \mathcal{L}(x_t, \eta_t))^2\right] \leq \mathcal{O}\left(\frac{q}{N_1} + \frac{q^2\epsilon^2}{N_2} + \frac{q}{N_2}\sum_{t=t_0}^{t_0+q-1}(g_t)^2\right), \tag{17}$$

$$\sum_{t=t_0}^{t_0+q-1} \mathbb{E}[\|v_t - \nabla_x \mathcal{L}(x_t, \eta_{t+1})\|^2] \leq \mathcal{O}\left(\frac{q}{N_3} + \frac{q^2\epsilon^2}{N_4} + \left(\frac{q}{N_3} + \frac{q}{N_4} + \frac{q^2\epsilon^2}{N_2N_4}\right)\sum_{t=t_0}^{t_0+q-1}(g_t)^2\right). \tag{18}$$

The full version and its proof can be found in Appendix A.9. The key idea is as follows. From the update of $g_t$, if $t \mod q \neq 0$, we can show that

$$\mathbb{E}[(g_t - \nabla_\eta \mathcal{L}(x_t, \eta_t))^2] = \mathbb{E}[(\nabla_\eta \mathcal{L}(x_t, \eta_t, \mathcal{B}_2) - \nabla_\eta \mathcal{L}(x_{t-1}, \eta_{t-1}, \mathcal{B}_2) + \nabla_\eta \mathcal{L}(x_{t-1}, \eta_{t-1})$$
$$- \nabla_\eta \mathcal{L}(x_t, \eta_t))^2] + \mathbb{E}[(g_{t-1} - \nabla_\eta \mathcal{L}(x_{t-1}, \eta_{t-1}))^2]. \tag{19}$$

The first term in RHS of equation 19 corresponds to the variance of an unbiased estimate of $\nabla_\eta \mathcal{L}(x_{t-1}, \eta_{t-1}) - \nabla_\eta \mathcal{L}(x_t, \eta_t)$ using $N_2$ samples. This term can be further bounded by $\mathbb{E}[(\nabla_\eta \mathcal{L}(x_t, \eta_t, \mathcal{B}_2) - \nabla_\eta \mathcal{L}(x_{t-1}, \eta_{t-1}, \mathcal{B}_2))^2]$, which is small due to our designed step sizes and Lemma 4. This bounds the first term in RHS of equation 19. Then, applying equation 19 recursively, we get the bound in equation 17. This method applies to $v_t$ too.

From the descent lemma and Lemma 2, similar to the previous proof in Theorem 3, we can show that $\sum_{t=0}^{T-1} \mathbb{E}[\alpha_t(g_t)^2 + \beta_t\|v_t\|^2]$ is bounded by the mean square errors of $g_t$ and $v_t$ plus some constants.

Using Lemma 5 where the mean square errors are upper bounded by a function of $\sum_{t=0}^{T-1} |g_t|^2$ and $\alpha_t = \frac{1}{4L_2}$, we can show that $\sum_{t=0}^{T-1} \mathbb{E}[(g_t)^2 + \beta_t \|v_t\|^2]$ is bounded by some constants. We then can show that for some $t < T$, $\mathbb{E}[(g_t)^2]$ is bounded. Moreover, with $\mathbb{E}[(g_t)^2]$ bounded, we can further show the mean square errors of $g_t$ and $v_t$ are bounded. We then obtain an $\epsilon$-stationary point.

## 4 NUMERICAL RESULTS

In this section, we conduct numerical studies on a set of regression tasks (Chen et al., 2023) on the life expectancy data [1]. This dataset consists of $N = 2413$ samples, where we select the first 2000 samples for training and the rest samples for testing. After removing redundant features, for the $i$-th sample in our dataset, we have input $z_i \in \mathbb{R}^{34}$ and output $y_i \in \mathbb{R}$. The non-convex original loss function is set as $\ell(x, (z_i, y_i)) = \frac{1}{2}(y_i - z_i^\top x)^2 + 0.1 \sum_{j=1}^{34} \ln(1 + |x_j|)$, where $x = (x_1, x_2, ..., x_{34})$ is the trainable parameter. For the DRO model, $\lambda$ is set to 0.01, and the initial value $\eta_0$ is set to 0.1.

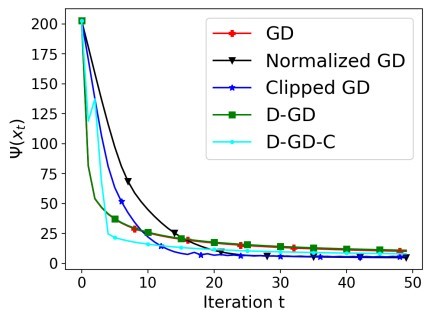
Figure 1: Deterministic setting.

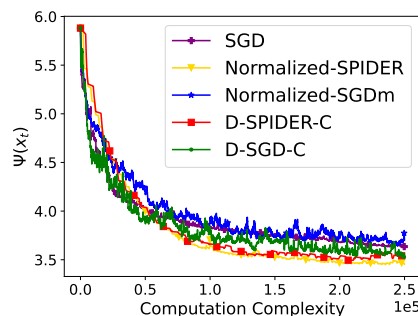
Figure 2: Stochastic setting.

In Figure 1, we provide the training curves with fine-tuned learning rate for the original GD, Normalized GD (Chen et al., 2023), Clipped GD (Zhang et al., 2019) and our proposed D-GD, D-GD-C methods. The x-axis stands for the training iteration, and the y-axis stands for the DRO objective $\psi(x)$. From the results of GD and D-GD in Figure 1, we can show that updating $x$ and $\eta$ alternatively has the same empirical performance compared with training them together. Our D-GD-C also has similar performance compared with other methods.

In Figure 2, we provide the training curves with fine-tuned learning rate for SGD, Normalized-SPIDER (Chen et al., 2023), Normalized-SGD with momentum (Jin et al., 2021) and our proposed D-SGD-C and D-SPIDER-C methods. Our D-SPIDER-C has similar performance compared with Normalized-SPIDER and both our two algorithms outperform the SGD and Normalized-SGD with momentum methods. The details can be found in Appendix A.10.

## 5 CONCLUSION

In this paper, we revisit the DRO problem with non-convex smooth loss functions. Instead of solving this problem as a generalized $(L_0, L_1)$-smoothness problem, we first show that this DRO problem satisfies a simpler yet more precise partially generalized smoothness condition and partially generalized affine noise condition. Under these conditions, our theoretical analyses are much simpler than existing studies. Our results provide new insights into the fundamental structure of DRO problems with non-convex loss functions, which could be useful for DRO problems beyond the one studied in this paper.

---

[1]https://www.kaggle.com/datasets/kumarajarshi/life-expectancy-who?resource=download

## 6 ACKNOWLEDGMENTS

The work of Q. Zhang and S. Zou was partially supported by NSF under Grant CCF-2438429. Y. Zhou's work was supported by the National Science Foundation under grants DMS-2134223 and ECCS-2237830. L. Shen's work was supported by the National Science Foundation under grant DMS-2208385. This work is also partially supported by the AFRL VFRP program.

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

## A APPENDIX

### A.1 PROOF OF LEMMA 1

*Proof.* Based on the formulation in equation 6 and the fact that $\mathcal{L}(x, \eta) = \hat{\mathcal{L}}(x, G\eta)$, we first have that

$$\nabla \mathcal{L}(x, \eta) = \left[ \mathbb{E}_{P_0} \left[ (\psi^*)' \left( \frac{\ell(x, S) - G\eta}{\lambda} \right) \nabla \ell(x, S) \right]; \right.$$

$$\left. G - G\mathbb{E}_{P_0} \left[ (\psi^*)' \left( \frac{\ell(x, S) - G\eta}{\lambda} \right) \right] \right]. \tag{20}$$

We first prove that the function $\nabla \mathcal{L}(x, \eta)$ is $L_2$-smooth in $\eta$. For any $x \in \mathbb{R}^d$ and $\eta, \eta' \in \mathbb{R}$, we have that

$$|\nabla_\eta \mathcal{L}(x, \eta) - \nabla_\eta \mathcal{L}(x, \eta')|$$

$$= G \left| \mathbb{E}_{P_0} \left[ (\psi^*)' \left( \frac{\ell(x, S) - G\eta}{\lambda} \right) - (\psi^*)' \left( \frac{\ell(x, S) - G\eta'}{\lambda} \right) \right] \right|$$

$$\leq G \mathbb{E}_{P_0} \left[ \left| (\psi^*)' \left( \frac{\ell(x, S) - G\eta}{\lambda} \right) - (\psi^*)' \left( \frac{\ell(x, S) - G\eta'}{\lambda} \right) \right| \right]$$

$$\leq \frac{G^2 M}{\lambda} |\eta - \eta'|, \tag{21}$$

where the inequality is due to the $M$-smoothness of the conjugate function. We further show the partially generalized smoothness in $x$. For any $x, x' \in \mathbb{R}^d$ and $\eta \in \mathbb{R}$, we first have $(\psi^*)' \left( \frac{\ell(x', S) - G\eta}{\lambda} \right) \geq 0$. This is due to the fact that $(\psi^*)'$ is monotonically increasing and $0 \leq \lim_{a \to -\infty} (\psi^*)'(a) \leq 1$, where the details can be found in Proposition B.2 in Jin et al. (2021). It then follows that

$$\|\nabla_x \mathcal{L}(x, \eta) - \nabla_x \mathcal{L}(x', \eta)\|$$

$$= \left\| \mathbb{E}_{P_0} \left[ (\psi^*)' \left( \frac{\ell(x, S) - G\eta}{\lambda} \right) \nabla \ell(x, S) - (\psi^*)' \left( \frac{\ell(x', S) - G\eta}{\lambda} \right) \nabla \ell(x', S) \right] \right\|$$

$$\leq \left\| \mathbb{E}_{P_0} \left[ (\psi^*)' \left( \frac{\ell(x, S) - G\eta}{\lambda} \right) \nabla \ell(x, S) - (\psi^*)' \left( \frac{\ell(x, S) - G\eta}{\lambda} \right) \nabla \ell(x', S) \right] \right\|$$

$$+ \left\| \mathbb{E}_{P_0} \left[ (\psi^*)' \left( \frac{\ell(x, S) - G\eta}{\lambda} \right) \nabla \ell(x', S) - (\psi^*)' \left( \frac{\ell(x', S) - G\eta}{\lambda} \right) \nabla \ell(x', S) \right] \right\|$$

$$\leq L \|x - x'\| \mathbb{E}_{P_0} \left[ (\psi^*)' \left( \frac{\ell(x, S) - G\eta}{\lambda} \right) \right] + GM \frac{G}{\lambda} \|x - x'\|$$

$$\leq \frac{L}{G} (|\nabla_\eta \mathcal{L}(x, \eta)| + G) \|x - x'\| + GM \frac{G}{\lambda} \|x - x'\|$$

$$\leq (L_0 + L_1 |\nabla_\eta \mathcal{L}(x, \eta)|) \|x - x'\|, \tag{22}$$

where the second inequality is because that $\ell(x, s)$ is $L$-smooth and $G$-continuous in $x$ for any $s \in \mathbb{S}$, and the conjugate function $\psi^*$ is $M$-smooth, the third inequality is due to the fact that $(\psi^*)' \left( \frac{\ell(x', S) - G\eta}{\lambda} \right) \geq 0$, and the fourth inequality is because that $G\mathbb{E}_{P_0} \left[ (\psi^*)' \left( \frac{\ell(x, S) - G\eta}{\lambda} \right) \right] = -\nabla_\eta \mathcal{L}(x, \eta) + G \leq |\nabla_\eta \mathcal{L}(x, \eta)| + G$. This completes the proof. □

### A.2 PROOF OF THEOREM 1

*Proof.* Let $H = 2L_2(\mathcal{L}(x_0, \eta_0) - \inf_{x, \eta} \mathcal{L}(x, \eta))$ and $L' = L_0 + L_1 \sqrt{H}$. In the following, we show that under our selected step sizes, $|\nabla_\eta \mathcal{L}(x_t, \eta_{t+1})|$ is upper bounded by $\sqrt{H}$ for all $t \geq 0$ by induction.

**Base Case:** for $t = 0$, since $\mathcal{L}(x, \eta)$ is $L_2$-smooth in $\eta$, from the descent lemma for standard $L$-smooth function we can show that

$$\mathcal{L}(x_0, \eta_1) \leq \mathcal{L}(x_0, \eta_0) - \langle \nabla_\eta \mathcal{L}(x_0, \eta_0), \alpha_0 \nabla_\eta \mathcal{L}(x_0, \eta_0) \rangle + \frac{L_2}{2} (\alpha_0 \nabla_\eta \mathcal{L}(x_0, \eta_0))^2. \tag{23}$$

Since $\alpha_0 = \frac{1}{L_2}$, it follows that

$$\frac{1}{2L_2}(\nabla_\eta \mathcal{L}(x_0, \eta_0))^2 \leq \mathcal{L}(x_0, \eta_0) - \mathcal{L}(x_0, \eta_1). \tag{24}$$

By Lemma 3.5 in Li et al. (2024), it can be shown that

$$(\nabla_\eta \mathcal{L}(x_0, \eta_1))^2 \leq 2L_2 \left( \mathcal{L}(x_0, \eta_1) - \inf_\eta \mathcal{L}(x_0, \eta) \right) \leq 2L_2 \left( L(x_0, \eta_1) - \inf_{x,\eta} \mathcal{L}(x, \eta) \right). \tag{25}$$

According to equation 24 and equation 25, we can show that $L(x_0, \eta_1) \leq L(x_0, \eta_0)$ and $|\nabla_\eta \mathcal{L}(x_0, \eta_1)| \leq \sqrt{H}$.

**Inductive Step:** Assume $|\nabla_\eta \mathcal{L}(x_t, \eta_{t+1})| \leq \sqrt{H}$ holds for $0 \leq t \leq t'$. In the following, we show that $|\nabla_\eta \mathcal{L}(x_t, \eta_{t+1})| \leq \sqrt{H}$ holds for $t = t' + 1$.

Due to the $L_2$-smoothness of $\mathcal{L}(x, \eta)$ in $\eta$ and $\alpha_t = \frac{1}{L_2}$, it can be shown that

$$\mathcal{L}(x_t, \eta_{t+1}) \leq \mathcal{L}(x_t, \eta_t) - \langle \nabla_\eta \mathcal{L}(x_t, \eta_t), \alpha_t \nabla_\eta \mathcal{L}(x_t, \eta_t) \rangle + \frac{L_2}{2}(\alpha_t \nabla_\eta \mathcal{L}(x_t, \eta_t))^2 \tag{26}$$

and

$$\frac{1}{2L_2}(\nabla_\eta \mathcal{L}(x_t, \eta_t))^2 \leq \mathcal{L}(x_t, \eta_t) - \mathcal{L}(x_t, \eta_{t+1}) \tag{27}$$

hold for all $t \geq 0$. Based on Lemma 2, we have that

$$\begin{aligned}
\mathcal{L}(x_{t+1}, \eta_{t+1}) \leq & \mathcal{L}(x_t, \eta_{t+1}) - \langle \nabla_x \mathcal{L}(x_t, \eta_{t+1}), \beta_t \nabla_x \mathcal{L}(x_t, \eta_{t+1}) \rangle \\
& + \frac{L_0 + L_1 |\nabla_\eta \mathcal{L}(x_t, \eta_{t+1})|}{2} \|\beta_t \nabla_x \mathcal{L}(x_t, \eta_{t+1})\|^2 \\
\leq & \mathcal{L}(x_t, \eta_{t+1}) - \langle \nabla_x \mathcal{L}(x_t, \eta_{t+1}), \beta_t \nabla_x \mathcal{L}(x_t, \eta_{t+1}) \rangle \\
& + \frac{L_0 + L_1 \sqrt{H}}{2} \|\beta_t \nabla_x \mathcal{L}(x_t, \eta_{t+1})\|^2
\end{aligned}$$

holds for all $0 \leq t \leq t'$. By setting $\beta_t = \frac{1}{L_0 + L_1\sqrt{H}} = \frac{1}{L'}$, it can further imply that

$$\frac{1}{2L'}\|\nabla_x \mathcal{L}(x_t, \eta_{t+1})\|^2 \leq \mathcal{L}(x_t, \eta_{t+1}) - \mathcal{L}(x_{t+1}, \eta_{t+1}) \tag{28}$$

holds for all $0 \leq t \leq t'$. By taking the sum of equation 27 from $t = 0$ to $t' + 1$ and equation 28 from $t = 0$ to $t'$, we can show that

$$0 \leq \sum_{t=0}^{t'+1} \frac{1}{2L_2}(\nabla_\eta \mathcal{L}(x_t, \eta_t))^2 + \sum_{t=0}^{t'} \frac{1}{2L'}\|\nabla_x \mathcal{L}(x_t, \eta_{t+1})\|^2 \leq \mathcal{L}(x_0, \eta_0) - \mathcal{L}(x_{t'+1}, \eta_{t'+2}).$$

By Lemma 3.5 in Li et al. (2024), it follows that

$$\begin{aligned}
(\nabla_\eta \mathcal{L}(x_{t'+1}, \eta_{t'+2}))^2 \leq & 2L_2 \left( \mathcal{L}(x_{t'+1}, \eta_{t'+2}) - \inf_\eta \mathcal{L}(x_{t'+1}, \eta) \right) \\
\leq & 2L_2 \left( L(x_0, \eta_1) - \inf_{x,\eta} \mathcal{L}(x, \eta) \right) = H,
\end{aligned}$$

which completes the inductive step. Since both the base case and the inductive step are proven, by the principle of mathematical induction, $|\nabla_\eta \mathcal{L}(x_t, \eta_{t+1})| \leq \sqrt{H}$ holds for any $t \geq 0$. This further implies that equation 28 holds for any $t \geq 0$. Then, we take the sum of equation 27 and equation 28 from $t = 0$ to $T - 1$, and show that

$$\sum_{t=0}^{T-1} \left( (\nabla_\eta \mathcal{L}(x_t, \eta_t))^2 + \|\nabla_x \mathcal{L}(x_t, \eta_{t+1})\|^2 \right) \leq \max(2L_2, 2L')(\mathcal{L}(x_0, \eta_0) - \mathcal{L}(x_T, \eta_T))$$

$$\leq \max(2L_2, 2L') \left( \mathcal{L}(x_0, \eta_0) - \inf_{x,\eta} \mathcal{L}(x, \eta) \right). \tag{29}$$

Thus for $T \geq 8 \max(L_2, L') \frac{\mathcal{L}(x_0, \eta_0) - \inf_{x,\eta} \mathcal{L}(x,\eta)}{\epsilon^2}$, we can find some $t$ such that

$$(\nabla_\eta \mathcal{L}(x_t, \eta_t))^2 + \|\nabla_x \mathcal{L}(x_t, \eta_{t+1})\|^2 \leq \frac{1}{4}\epsilon^2. \tag{30}$$

Thus we have $\|\nabla_x \mathcal{L}(x_t, \eta_{t+1})\| \leq \frac{\epsilon}{2}$ and $|\nabla_\eta \mathcal{L}(x_t, \eta_t)| \leq \frac{\epsilon}{2}$. Moreover, we can show that

$$\begin{aligned}
|\nabla_\eta \mathcal{L}(x_t, \eta_{t+1})| \leq & |\nabla_\eta \mathcal{L}(x_t, \eta_t)| + |\nabla_\eta \mathcal{L}(x_t, \eta_{t+1}) - \nabla_\eta \mathcal{L}(x_t, \eta_t)| \\
\leq & (1 + L_2 \alpha_t)|\nabla_\eta \mathcal{L}(x_t, \eta_t)| \leq \epsilon,
\end{aligned} \tag{31}$$

where the second inequality is due to the standard $L_2$-smoothness on $\eta$. This completes the proof. □

## A.3 Proof of Lemma 2

*Proof.* From Taylor's Theorem, we have that

$$\mathcal{L}(x', \eta) - \mathcal{L}(x, \eta) = \int_0^1 \langle \nabla_x \mathcal{L}(x + \theta(x' - x), \eta), x' - x \rangle d\theta. \tag{32}$$

It follows that

$$\begin{aligned}
& \mathcal{L}(x', \eta) - \mathcal{L}(x, \eta) - \langle \nabla_x \mathcal{L}(x, \eta), x' - x \rangle \\
= & \int_0^1 \langle \nabla_x \mathcal{L}(x + \theta(x' - x), \eta) - \nabla_x \mathcal{L}(x, \eta), x' - x \rangle d\theta \\
\leq & \int_0^1 (L_0 + L_1 |\nabla_\eta \mathcal{L}(x, \eta)|) \theta \|x' - x\|^2 d\theta \\
\leq & \frac{L_0 + L_1 |\nabla_\eta \mathcal{L}(x, \eta)|}{2} \|x' - x\|^2,
\end{aligned} \tag{33}$$

where the first inequality is due to the partially generalized smoothness condition and this completes the proof. □

## A.4 Proof of Theorem 2

*Proof.* Since $\mathcal{L}(x, \eta)$ is $L_2$-smooth in $\eta$, we have that

$$\mathcal{L}(x_t, \eta_{t+1}) \leq \mathcal{L}(x_t, \eta_t) - \langle \nabla_\eta \mathcal{L}(x_t, \eta_t), \alpha_t \nabla_\eta \mathcal{L}(x_t, \eta_t) \rangle + \frac{L_2}{2}(\alpha_t \nabla_\eta \mathcal{L}(x_t, \eta_t))^2.$$

For $\alpha_t = \frac{1}{L_2}$, we have that

$$\frac{\alpha_t}{2}(\nabla_\eta \mathcal{L}(x_t, \eta_t))^2 \leq \mathcal{L}(x_t, \eta_t) - \mathcal{L}(x_t, \eta_{t+1}). \tag{34}$$

For the update of $x$, by Lemma 2 we have that

$$\begin{aligned}
\mathcal{L}(x_{t+1}, \eta_{t+1}) \leq & \mathcal{L}(x_t, \eta_{t+1}) - \langle \nabla_x \mathcal{L}(x_t, \eta_{t+1}), \beta_t \nabla_x \mathcal{L}(x_t, \eta_{t+1}) \rangle \\
& + \frac{L_0 + L_1 |\nabla_\eta \mathcal{L}(x_t, \eta_{t+1})|}{2} \|\beta_t \nabla_x \mathcal{L}(x_t, \eta_{t+1})\|^2.
\end{aligned} \tag{35}$$

For $\beta_t = \min\left(\frac{1}{2L_0}, \frac{1}{2L_1 |\nabla_\eta \mathcal{L}(x_t, \eta_{t+1})|}\right) \leq \frac{1}{L_0 + L_1 |\nabla_\eta \mathcal{L}(x_t, \eta_{t+1})|}$, from equation 35, it follows that

$$\frac{\beta_t}{2}\|\nabla_x \mathcal{L}(x_t, \eta_{t+1})\|^2 \leq \mathcal{L}(x_t, \eta_{t+1}) - \mathcal{L}(x_{t+1}, \eta_{t+1}). \tag{36}$$

By taking the sum of equation 34 and equation 36 from $t = 0$ to $T - 1$, we then have that

$$\begin{aligned}
\sum_{t=0}^{T-1} \left(\frac{\alpha_t}{2}(\nabla_\eta \mathcal{L}(x_t, \eta_t))^2 + \frac{\beta_t}{2}\|\nabla_x \mathcal{L}(x_t, \eta_{t+1})\|^2\right) \leq & \mathcal{L}(x_0, \eta_0) - \mathcal{L}(x_T, \eta_T) \\
\leq & \mathcal{L}(x_0, \eta_0) - \inf_{x,\eta} \mathcal{L}(x, \eta).
\end{aligned} \tag{37}$$

For $T \geq \frac{\mathcal{L}(x_0, \eta_0) - \inf_{x,\eta} \mathcal{L}(x,\eta)}{\epsilon^2} \max(8L_2, 16L_0)$, we can find some $t < T$ such that

$$\alpha_t (\nabla_\eta \mathcal{L}(x_t, \eta_t))^2 + \beta_t \|\nabla_x \mathcal{L}(x_t, \eta_{t+1})\|^2 \leq \frac{\epsilon^2}{4} \min\left(\frac{1}{L_2}, \frac{1}{2L_0}\right). \tag{38}$$

For this $t$, we have that

$$\alpha_t (\nabla_\eta \mathcal{L}(x_t, \eta_t))^2 \leq \frac{\epsilon^2}{4L_2},$$

which leads to $|\nabla_\eta \mathcal{L}(x_t, \eta_t)| \leq \frac{\epsilon}{2}$ and

$$\begin{aligned}
|\nabla_\eta \mathcal{L}(x_t, \eta_{t+1})| &\leq |\nabla_\eta \mathcal{L}(x_t, \eta_t)| + |\nabla_\eta \mathcal{L}(x_t, \eta_{t+1}) - \nabla_\eta \mathcal{L}(x_t, \eta_t)| \\
&\leq (1 + L_2 \alpha_t)|\nabla_\eta \mathcal{L}(x_t, \eta_t)| \leq \epsilon.
\end{aligned} \tag{39}$$

Moreover, since $|\nabla_\eta \mathcal{L}(x_t, \eta_{t+1})| \leq \epsilon \leq \frac{L_0}{L_1}$, we have that $\beta_t = \min\left(\frac{1}{2L_0}, \frac{1}{2L_1 |\nabla_\eta \mathcal{L}(x_t, \eta_{t+1})|}\right) = \frac{1}{2L_0}$. It follows that

$$\frac{1}{2L_0} \|\nabla_x \mathcal{L}(x_t, \eta_{t+1})\|^2 \leq \frac{1}{2L_0} \frac{\epsilon^2}{4}, \tag{40}$$

which implies that $\|\nabla_x \mathcal{L}(x_t, \eta_{t+1})\| \leq \frac{\epsilon}{2}$. This completes the proof. $\qquad \square$

## A.5 PROOF OF LEMMA 3

*Proof.* We first consider the variance of the gradient to $\eta$. We first have that

$$\begin{aligned}
&\mathbb{V}_{S \sim P_0}[\nabla_\eta \mathcal{L}(x, \eta, S)] \\
=&\mathbb{E}_{S \sim P_0}[(\nabla_\eta \mathcal{L}(x, \eta, S) - \nabla_\eta \mathcal{L}(x, \eta))^2] \\
=&\frac{1}{2} \mathbb{E}_{S_1, S_2 \sim P_0}[(\nabla_\eta \mathcal{L}(x, \eta, S_1) - \nabla_\eta \mathcal{L}(x, \eta, S_2))^2] \\
=&\frac{1}{2} G^2 \mathbb{E}_{S_1, S_2 \sim P_0}\left[\left((\psi^*)'\left(\frac{\ell(x, S_1) - G\eta}{\lambda}\right) - (\psi^*)'\left(\frac{\ell(x, S_1) - G\eta}{\lambda}\right)\right)^2\right] \\
\leq& G^2 M^2 \lambda^{-2} \mathbb{V}_{S \sim P_0}[\ell(x, S)] \\
\leq& G^2 M^2 \lambda^{-2} \sigma^2 = D_2,
\end{aligned} \tag{41}$$

where the first inequality is due to the $M$-smoothness in $\eta$, $S_1, S_2$ are independent random variables with the same distribution $P_0$ and the last inequality is due to Assumption 3. We then provide the proof for the variance of the gradient to $x$. Similarly, we have that

$$\begin{aligned}
&\mathbb{V}_{S \sim P_0}[\nabla_x \mathcal{L}(x, \eta, S)] \\
=&\frac{1}{2} \mathbb{E}_{S_1, S_2 \sim P_0}[\|\nabla_x \mathcal{L}(x, \eta, S_1) - \nabla_x \mathcal{L}(x, \eta, S_2)\|^2] \\
\leq&\mathbb{E}_{S_1, S_2 \sim P_0}\left[\left\|(\psi^*)'\left(\frac{\ell(x, S_1) - G\eta}{\lambda}\right)^2 (\nabla \ell(x, S_1) - \nabla \ell(x, S_2))\right\|^2\right] \\
&+ \mathbb{E}_{S_1, S_2 \sim P_0}\left[\left((\psi^*)'\left(\frac{\ell(x, S_1) - G\eta}{\lambda}\right) - (\psi^*)'\left(\frac{\ell(x, S_2) - G\eta}{\lambda}\right)\right)^2 \|\nabla \ell(x, S_2)\|^2\right] \\
\leq&4G^2 \mathbb{E}_{S_1 \sim P_0}\left[\left((\psi^*)'\left(\frac{\ell(x, S_1) - G\eta}{\lambda}\right)\right)^2\right] + 2G^2 M^2 \lambda^{-2} \sigma^2 \\
\leq&8(G^2 + |\nabla_\eta \mathcal{L}(x, \eta)|^2 + D_2) + 2G^2 M^2 \lambda^{-2} \sigma^2 \\
\leq&8G^2 + 10G^2 M^2 \lambda^{-2} \sigma^2 + 8|\nabla_\eta \mathcal{L}(x, \eta)|^2,
\end{aligned} \tag{42}$$

where the second inequality is due to the $G$-continuous property of $\ell$, $M$-smoothness of $\psi^*$ and Assumption 3. The third inequality is due to the fact that $\mathbb{E}_{S_1 \sim P_0}\left[\left((\psi^*)'\left(\frac{\ell(x, S_1) - G\eta}{\lambda}\right)\right)^2\right] = \left(\mathbb{E}_{S_1 \sim P_0}\left[(\psi^*)'\left(\frac{\ell(x, S_1) - G\eta}{\lambda}\right)\right]\right)^2 + \mathbb{V}_{S_1 \sim P_0}\left[(\psi^*)'\left(\frac{\ell(x, S_1) - G\eta}{\lambda}\right)\right]$. This completes the proof. $\quad \square$

## A.6 Full version of Theorem 3 and its proof

**Theorem 5.** *Let* $\alpha = \frac{1}{2L_2}, \beta_t = \min\left(\frac{1}{2L_0}, \frac{\epsilon}{L_0\|v_t\|}\right)$ *and* $\gamma = \frac{1}{36L_1^2 L_2}$. *Setting* $T \geq \max(8L_2, 4L_0)\frac{5\mathcal{L}(x_0,\eta_0)-5\inf_{x,\eta}\mathcal{L}(x,\eta)}{\epsilon^2}$, $N_1 \geq \max\left(\frac{20D_2}{\epsilon^2}, \frac{10D_2 L_0}{L_2\epsilon^2}\right)$, $N_2 \geq \max\left(\frac{10D_0 L_2}{L_0\epsilon^2}, \frac{5D_0}{\epsilon^2}, \frac{12D_1 L_2}{L_0}, \frac{D_0+4\epsilon^2 D_1}{\epsilon^2}\right)$ *and* $\epsilon^2 \leq \frac{8\gamma L_0^4}{5}\min\left(\frac{1}{8L_2}, \frac{1}{4L_0}\right)$, *we have that*

$$\min_{t<T}\mathbb{E}[\|\nabla_{x,\eta}\mathcal{L}(x_t,\eta_{t+1})\|] \leq 6\epsilon.$$

*Proof.* Since $\mathcal{L}(x,\eta)$ is $L_2$-smooth in $\eta$, we have that

$$\mathcal{L}(x_t,\eta_{t+1}) \leq \mathcal{L}(x_t,\eta_t) - \langle\nabla_\eta\mathcal{L}(x_t,\eta_t),\alpha_t g_t\rangle + \frac{L_2}{2}(\alpha_t g_t)^2$$
$$\leq \mathcal{L}(x_t,\eta_t) - \langle\nabla_\eta\mathcal{L}(x_t,\eta_t),\alpha_t g_t\rangle + \alpha_t^2 L_2((\nabla_\eta\mathcal{L}(x_t,\eta_t))^2 + (g_t - \nabla_\eta\mathcal{L}(x_t,\eta_t))^2).$$

Taking the expectation on both sides of the above inequality, we can further show that

$$\mathbb{E}[\mathcal{L}(x_t,\eta_{t+1})] \leq \mathbb{E}[\mathcal{L}(x_t,\eta_t)] - \mathbb{E}[\alpha_t(\nabla_\eta\mathcal{L}(x_t,\eta_t))^2]$$
$$+ \mathbb{E}[\alpha_t^2 L_2((\nabla_\eta\mathcal{L}(x_t,\eta_t))^2 + (g_t - \nabla_\eta\mathcal{L}(x_t,\eta_t))^2)], \tag{43}$$

since $g_t$ is an unbiased estimate of $\nabla_\eta\mathcal{L}(x_t,\eta_t)$. According to Lemma 2, for any $\gamma > 0$, we have that

$$\mathcal{L}(x_{t+1},\eta_{t+1}) \leq \mathcal{L}(x_t,\eta_{t+1}) - \langle\nabla_x\mathcal{L}(x_t,\eta_{t+1}),\beta_t v_t\rangle + \frac{L_0 + L_1|\nabla_\eta\mathcal{L}(x_t,\eta_{t+1})|}{2}\|\beta_t v_t\|^2$$
$$\leq \mathcal{L}(x_t,\eta_{t+1}) - \langle\nabla_x\mathcal{L}(x_t,\eta_{t+1}),\beta_t v_t\rangle$$
$$+ \frac{L_0 + L_1|\nabla_\eta\mathcal{L}(x_t,\eta_t)| + L_1 L_2|\alpha_t g_t|}{2}\|\beta_t v_t\|^2$$
$$\leq \mathcal{L}(x_t,\eta_{t+1}) - \frac{\beta_t}{2}(\|v_t\|^2 + \|\nabla_x\mathcal{L}(x_t,\eta_{t+1})\|^2 - \|v_t - \nabla_x\mathcal{L}(x_t,\eta_{t+1})\|^2)$$
$$+ \frac{L_0 + (L_1 + L_1 L_2\alpha_t)|\nabla_\eta\mathcal{L}(x_t,\eta_t)| + L_1 L_2\alpha_t|g_t - \nabla_\eta\mathcal{L}(x_t,\eta_t)|}{2}\|\beta_t v_t\|^2$$
$$\leq \mathcal{L}(x_t,\eta_{t+1}) - \frac{\beta_t}{2}\|v_t\|^2 + \frac{\beta_t}{2}\|v_t - \nabla_x\mathcal{L}(x_t,\eta_{t+1})\|^2 + \frac{L_0}{2}\beta_t^2\|v_t\|^2$$
$$+ \gamma(L_1 + L_1 L_2\alpha_t)^2(\nabla_\eta\mathcal{L}(x_t,\eta_t))^2 + \frac{\beta_t^4}{16\gamma}\|v_t\|^4$$
$$+ \gamma(L_1 L_2\alpha_t)^2(g_t - \nabla_\eta\mathcal{L}(x_t,\eta_t))^2 + \frac{\beta_t^4}{16\gamma}\|v_t\|^4, \tag{44}$$

where the second inequality is due to the $L_2$-smoothness in $\eta$, and the last inequality is due to that $2ab \leq a^2 + b^2$ for any $a,b \in \mathbb{R}$. Taking the expectation on both sides of equation 44 and adding with equation 43, it follows that

$$\mathbb{E}\left[\left(\alpha_t - L_2\alpha_t^2 - \gamma(L_1 + L_1 L_2\alpha_t)^2\right)(\nabla_\eta\mathcal{L}(x_t,\eta_t))^2 + \left(\frac{\beta_t}{2} - \frac{L_0\beta_t^2}{2}\right)\|v_t\|^2\right]$$
$$\leq \mathbb{E}[\mathcal{L}(x_t,\eta_t) - \mathcal{L}(x_{t+1},\eta_{t+1})] + \mathbb{E}[(\alpha_t^2 L_2 + \gamma(L_1 L_2\alpha_t)^2)(g_t - \nabla_\eta\mathcal{L}(x_t,\eta_t))^2]$$
$$+ \mathbb{E}\left[\frac{\beta_t}{2}\|v_t - \nabla_x\mathcal{L}(x_t,\eta_{t+1})\|^2\right] + \mathbb{E}\left[\frac{\beta_t^4}{8\gamma}\|v_t\|^4\right]. \tag{45}$$

According to $\alpha = \frac{1}{2L_2}, \beta_t = \min\left(\frac{1}{2L_0}, \frac{\epsilon}{L_0\|v_t\|}\right)$ and Lemma 3, we can further show that

$$\mathbb{E}\left[\left(\frac{1}{4L_2} - \gamma\frac{9L_1^2}{4}\right)(\nabla_\eta\mathcal{L}(x_t,\eta_t))^2 + \frac{\beta_t}{4}\|v_t\|^2\right]$$
$$\leq \mathbb{E}[\mathcal{L}(x_t,\eta_t) - \mathcal{L}(x_{t+1},\eta_{t+1})] + \left(\frac{1}{4L_2} + \frac{\gamma L_1^2}{4}\right)\frac{D_2}{N_1}$$

$$+ \frac{1}{4L_0}\mathbb{E}\left[\frac{D_0 + D_1(\nabla_\eta\mathcal{L}(x_t, \eta_{t+1}))^2}{N_2}\right] + \frac{\epsilon^4}{8\gamma L_0^4}. \tag{46}$$

Moreover, we have that

$$\begin{aligned}
\mathbb{E}[(\nabla_\eta\mathcal{L}(x_t, \eta_{t+1}))^2] &\leq 2\mathbb{E}[(\nabla_\eta\mathcal{L}(x_t, \eta_t))^2] + 2\mathbb{E}[(\nabla_\eta\mathcal{L}(x_t, \eta_{t+1}) - \nabla_\eta\mathcal{L}(x_t, \eta_t))^2] \\
&\leq 2\mathbb{E}[(\nabla_\eta\mathcal{L}(x_t, \eta_t))^2] + 2L_2^2\alpha_t^2\mathbb{E}[(g_t)^2] \\
&\leq (2 + 4L_2^2\alpha_t^2)\mathbb{E}[(\nabla_\eta\mathcal{L}(x_t, \eta_t))^2] + 4L_2^2\alpha_t^2\mathbb{E}[(g_t - \nabla_\eta\mathcal{L}(x_t, \eta_t))^2] \\
&\leq 3\mathbb{E}[(\nabla_\eta\mathcal{L}(x_t, \eta_t))^2] + \frac{D_2}{N_1}. \tag{47}
\end{aligned}$$

Let $\gamma = \frac{1}{36L_1^2 L_2}$ and $N_2 \geq \frac{12D_1 L_2}{L_0}$. By equation 46 and equation 47, we then have that

$$\begin{aligned}
&\mathbb{E}\left[\frac{1}{8L_2}(\nabla_\eta\mathcal{L}(x_t, \eta_t))^2 + \frac{\beta_t}{4}\|v_t\|^2\right] \\
&\leq \mathbb{E}[\mathcal{L}(x_t, \eta_t) - \mathcal{L}(x_{t+1}, \eta_{t+1})] + \frac{D_2}{2L_2 N_1} + \frac{D_0}{4L_0 N_2} + \frac{D_1}{4L_0 N_2}\frac{D_2}{N_1} + \frac{\epsilon^4}{8\gamma L_0^4}. \tag{48}
\end{aligned}$$

Taking the sum of equation 48 from $t = 0$ to $T - 1$, we have that

$$\begin{aligned}
&\frac{1}{T}\sum_{t=0}^{T-1}\mathbb{E}\left[\frac{1}{8L_2}(\nabla_\eta\mathcal{L}(x_t, \eta_t))^2 + \frac{\beta_t}{4}\|v_t\|^2\right] \\
&\leq \frac{1}{T}\mathbb{E}[\mathcal{L}(x_0, \eta_0) - \mathcal{L}(x_T, \eta_T)] + \frac{D_2}{2L_2 N_1} + \frac{D_0}{4L_0 N_2} + \frac{D_1}{4L_0 N_2}\frac{D_2}{N_1} + \frac{\epsilon^4}{8\gamma L_0^4}. \tag{49}
\end{aligned}$$

Let $T \geq \max(8L_2, 4L_0)\frac{5\mathcal{L}(x_0, \eta_0) - 5\inf_{x,\eta}\mathcal{L}(x,\eta)}{\epsilon^2}$, $N_1 \geq \max\left(\frac{20D_2}{\epsilon^2}, \frac{10D_2 L_0}{L_2\epsilon^2}\right)$, $N_2 \geq \max\left(\frac{10D_0 L_2}{L_0\epsilon^2}, \frac{5D_0}{\epsilon^2}, \frac{12D_1 L_2}{L_0}\right)$ and $\epsilon^2 \leq \frac{8\gamma L_0^4}{5}\min\left(\frac{1}{8L_2}, \frac{1}{4L_0}\right)$. Then equation 49 can be further bounded as follows:

$$\frac{1}{T}\sum_{t=0}^{T-1}\mathbb{E}\left[\frac{1}{8L_2}(\nabla_\eta\mathcal{L}(x_t, \eta_t))^2 + \frac{\beta_t}{4}\|v_t\|^2\right] \leq \min\left(\frac{1}{8L_2}, \frac{1}{4L_0}\right)\epsilon^2. \tag{50}$$

Thus we can find some $t < T$ such that $\mathbb{E}\left[\frac{1}{8L_2}(\nabla_\eta\mathcal{L}(x_t, \eta_t))^2\right] \leq \frac{\epsilon^2}{8L_2}$ and $\mathbb{E}\left[\frac{\beta_t}{4}\|v_t\|^2\right] \leq \frac{1}{4L_0}\epsilon^2$.

Based on equation 47, we then have that $(\mathbb{E}[\nabla_\eta\mathcal{L}(x_t, \eta_{t+1})])^2 \leq \mathbb{E}[(\nabla_\eta\mathcal{L}(x_t, \eta_{t+1}))^2] \leq 4\epsilon^2$. Moreover, we have that

$$\frac{\epsilon^2}{L_0} \geq \mathbb{E}[\beta_t\|v_t\|^2] = \mathbb{E}\left[\frac{\epsilon^2}{L_0}\min\left(\frac{\|v_t\|^2}{2\epsilon^2}, \frac{\|v_t\|}{\epsilon}\right)\right] \geq \mathbb{E}\left[\frac{\epsilon\|v_t\|}{L_0}\right] - \frac{\epsilon^2}{2L_0}, \tag{51}$$

where the second inequality is due to the fact that $\min\left(\frac{a^2}{2}, a\right) \geq a - \frac{1}{2}$ holds for any $a \geq 0$. As a result, we have that $\mathbb{E}[\|v_t\|] \leq 3\epsilon$. For $N_2 \geq \frac{D_0 + 4\epsilon^2 D_1}{\epsilon^2}$, we have that

$$(\mathbb{E}[\|v_t - \nabla_x\mathcal{L}(x_t, \eta_{t+1})\|])^2 \leq \mathbb{E}[\|v_t - \nabla_x\mathcal{L}(x_t, \eta_{t+1})\|]^2 \leq \mathbb{E}\left[\frac{D_0 + D_1(\nabla_\eta\mathcal{L}(x_t, \eta_{t+1}))^2}{N_2}\right] \leq \epsilon^2.$$

It can be further shown that $\mathbb{E}[\|\nabla_x\mathcal{L}(x_t, \eta_{t+1})\|] \leq \mathbb{E}[\|v_t\|] + \mathbb{E}[\|v_t - \nabla_x\mathcal{L}(x_t, \eta_{t+1})\|] \leq 4\epsilon$ and $\mathbb{E}[\|\nabla_{x,\eta}\mathcal{L}(x_t, \eta_{t+1})\|] \leq \mathbb{E}[\|\nabla_\eta\mathcal{L}(x_t, \eta_{t+1})\|] + \mathbb{E}[\|\nabla_x\mathcal{L}(x_t, \eta_{t+1})\|] \leq 6\epsilon$. This completes the proof. $\qquad\square$

### A.7 PROOF OF LEMMA 4

*Proof.* For any $x, x' \in \mathbb{R}^d, \eta \in \mathbb{R}$ and $s \in \mathbb{S}$, we have that

$$\begin{aligned}
&|\nabla_\eta\mathcal{L}(x, \eta, s) - \nabla_\eta\mathcal{L}(x', \eta, s)| \\
&= G\left|(\psi^*)'\left(\frac{\ell(x, s) - G\eta}{\lambda}\right) - (\psi^*)'\left(\frac{\ell(x', s) - G\eta}{\lambda}\right)\right|
\end{aligned}$$

$$\leq \frac{G^2 M}{\lambda} \|x - x'\|, \tag{52}$$

where the inequality is due to that $\psi^*$ is $M$-smooth and $\ell(x, s)$ is $G$-continuous in $x$. Similarly, for any $x \in \mathbb{R}^d, \eta, \eta' \in \mathbb{R}$ and $s \in \mathbb{S}$, we have that

$$\begin{aligned}
&\|\nabla_x \mathcal{L}(x, \eta, s) - \nabla_x \mathcal{L}(x, \eta', s)\| \\
&= \left\| (\psi^*)' \left( \frac{\ell(x, s) - G\eta}{\lambda} \right) \nabla \ell(x, s) - (\psi^*)' \left( \frac{\ell(x, s) - G\eta'}{\lambda} \right) \nabla \ell(x, s) \right\| \\
&\leq G \left| (\psi^*)' \left( \frac{\ell(x, s) - G\eta}{\lambda} \right) - (\psi^*)' \left( \frac{\ell(x, s) - G\eta'}{\lambda} \right) \right| \\
&\leq \frac{G^2 M}{\lambda} |\eta - \eta'|, \tag{53}
\end{aligned}$$

where the first inequality is because that $\ell(x, s)$ is $G$-continuous in $x$ and the last one is due to the $M$-smoothness of $\psi^*$. $\qquad \square$

## A.8 FULL VERSION AND ITS PROOF OF THEOREM 4

Let $c_0 = \max(32L_2, 8L_0), c_1 = \left( 4 + \frac{8L_1^2 D_2}{L_0^2} + \frac{32L_1^2 D_2}{N_1 L_0^2} + \frac{16L_1^2 L_2}{5D_1 L_0^3} \right), c_2 = \max \left( \frac{1}{8L_2} + \frac{L_1}{50}, 1 \right),$
$c_3 = 1 + \frac{L_2}{40L_0} + \frac{L_0 D_1 + L_0 + 2L_0 L_2 D_2}{L_2} + \frac{33L_1^2}{5L_0 L_2} + \frac{L_0 D_1}{15L_2^2} + \frac{L_1^2}{2L_0 L_2^2}$ and $c_4 = \frac{17}{4} + \sqrt{c_3} + \sqrt{\frac{1}{60L_2}}.$

For $N_1 \geq \frac{6D_2 c_0 c_2}{\epsilon^2}$, $N_2 \geq \max \left( \frac{20q D_1 L_2}{L_0}, 20qc_2 L_2, \frac{12q L_2^2 c_0 c_2}{L_0^2}, q \right)$, $N_3 \geq \max \left( \frac{200 D_1 L_2}{L_0}, \frac{3c_0(D_0 + 4D_1 D_2)}{2L_0 \epsilon^2} \right)$, $N_4 \geq \max \left( \frac{5q L_2}{L_0}, \frac{6qc_1 c_0}{L_0} \right)$, and $\epsilon^2 \leq \min \left( \frac{L_0^3 N_4}{1480 L_1^2 L_2 q}, \frac{L_0^4}{42 L_1 c_0}, 1 \right)$, we have the following theorem:

**Theorem 6.** *Let* $\alpha = \frac{1}{4L_2}, \beta_t = \min \left( \frac{1}{2L_0}, \frac{\epsilon}{L_0 \|v_t\|} \right)$ *for Algorithm 3. Setting* $T \geq \frac{6c_0(\mathcal{L}(x_0, \eta_0) - \inf_{x,\eta} \mathcal{L}(x, \eta))}{\epsilon^2}$, *we have that*

$$\min_{t < T} \mathbb{E}[\|\nabla_{x,\eta} \mathcal{L}(x_t, \eta_{t+1})\|] \leq c_4 \epsilon.$$

*Proof.* Similar to equation 43, for the update of $\eta$, we have that

$$\begin{aligned}
\mathbb{E}[\mathcal{L}(x_t, \eta_{t+1})] &\leq \mathbb{E} \left[ \mathcal{L}(x_t, \eta_t) - \langle \nabla_\eta \mathcal{L}(x_t, \eta_t), \alpha_t g_t \rangle + \frac{L_2}{2} (\alpha_t g_t)^2 \right] \\
&\leq \mathbb{E}[\mathcal{L}(x_t, \eta_t)] - \frac{1}{2} \mathbb{E}[\alpha_t(g_t)^2] + \frac{1}{2} \mathbb{E}[\alpha_t(g_t - \nabla_\eta \mathcal{L}(x_t, \eta_t))^2] + \frac{L_2}{2} \mathbb{E}[\alpha_t^2(g_t)^2]. \tag{54}
\end{aligned}$$

Similar to equation 44, for the update of $x$ and any $\gamma > 0$, we have that

$$\begin{aligned}
\mathcal{L}(x_{t+1}, \eta_{t+1}) &\leq \mathcal{L}(x_t, \eta_{t+1}) - \langle \nabla_x \mathcal{L}(x_t, \eta_{t+1}), \beta_t v_t \rangle \\
&\quad + \frac{L_0 + L_1 |\nabla_\eta \mathcal{L}(x_t, \eta_t)| + L_1 L_2 |\alpha_t g_t|}{2} \|\beta_t v_t\|^2 \\
&\leq \mathcal{L}(x_t, \eta_{t+1}) - \frac{\beta_t}{2} (\|v_t\|^2 + \|\nabla_x \mathcal{L}(x_t, \eta_{t+1})\|^2 - \|v_t - \nabla_x \mathcal{L}(x_t, \eta_{t+1})\|^2) \\
&\quad + \frac{L_0 + L_1 |\nabla_\eta \mathcal{L}(x_t, \eta_t) - g_t| + (L_1 + L_1 L_2 \alpha_t) |g_t|}{2} \|\beta_t v_t\|^2 \\
&\leq \mathcal{L}(x_t, \eta_{t+1}) - \frac{\beta_t}{2} \|v_t\|^2 + \frac{\beta_t}{2} \|v_t - \nabla_x \mathcal{L}(x_t, \eta_{t+1})\|^2 + \frac{L_0}{2} \beta_t^2 \|v_t\|^2 \\
&\quad + \gamma(L_1 + L_1 L_2 \alpha_t)^2 (g_t)^2 + \frac{\beta_t^4}{16\gamma} \|v_t\|^4 \\
&\quad + \gamma L_1^2 |g_t - \nabla_\eta \mathcal{L}(x_t, \eta_t)|^2 + \frac{\beta_t^4}{16\gamma} \|v_t\|^4. \tag{55}
\end{aligned}$$

Combine equation 54 and equation 55 and it follows that

$$\mathbb{E}\left[\left(\frac{1}{2}\alpha_t - \frac{L_2\alpha_t^2}{2} - \gamma(L_1 + L_1 L_2 \alpha_t)^2\right)(g_t)^2 + \left(\frac{\beta_t}{2} - \frac{L_0\beta_t^2}{2}\right)\|v_t\|^2\right]$$

$$\leq \mathbb{E}[\mathcal{L}(x_t, \eta_t) - \mathcal{L}(x_{t+1}, \eta_{t+1}) + \frac{\beta_t^4}{8\gamma}\|v_t\|^4$$

$$+ \frac{\beta_t}{2}\|v_t - \nabla_x \mathcal{L}(x_t, \eta_{t+1})\|^2 + \frac{\alpha_t}{2}(g_t - \nabla_\eta \mathcal{L}(x_t, \eta_t))^2 + \gamma L_1^2(g_t - \nabla_\eta \mathcal{L}(x_t, \eta_t))^2].$$

Setting $\alpha = \frac{1}{4L_2}$ and $\beta_t = \min\left(\frac{1}{2L_0}, \frac{\epsilon}{L_0\|v_t\|}\right)$, we have that

$$\mathbb{E}[(\frac{3}{32L_2} - \gamma\frac{25L_1^2}{16})(g_t)^2 + \frac{\beta_t}{4}\|v_t\|^2]$$

$$\leq \mathbb{E}[\mathcal{L}(x_t, \eta_t) - \mathcal{L}(x_{t+1}, \eta_{t+1})] + \frac{\epsilon^4}{8\gamma L_0^4} + \frac{1}{4L_0}\mathbb{E}[\|v_t - \nabla_x \mathcal{L}(x_t, \eta_{t+1})\|^2]$$

$$+ \left(\frac{1}{8L_2} + \gamma L_1^2\right)\mathbb{E}[(g_t - \nabla_\eta \mathcal{L}(x_t, \eta_t))^2]. \tag{56}$$

Let $\gamma = \frac{1}{50L_1}$ and take the sum of equation 55 from $t = 0$ to $T - 1$. According to Lemma 5, we then have that

$$\sum_{t=0}^{T-1}\frac{1}{16L_2}\mathbb{E}[(g_t)^2] + \mathbb{E}[\frac{\beta_t}{4}\|v_t\|^2]$$

$$\leq \mathbb{E}[\mathcal{L}(x_0, \eta_0) - \mathcal{L}(x_T, \eta_T)] + \frac{7\epsilon^4 L_1 T}{L_0^4}$$

$$+ \frac{1}{4L_0}\sum_{t=0}^{T-1}\mathbb{E}\left[\frac{D_0}{N_3} + \frac{4D_1 D_2}{N_3 N_1} + \frac{q\epsilon^2}{N_4}\left(4 + \frac{8L_1^2 D_2}{L_0^2} + \frac{32L_1^2 D_2}{N_1 L_0^2} + \frac{64q\epsilon^2 L_1^2 L_2^2}{N_2 L_0^4}\right)\right]$$

$$+ \frac{1}{4L_0}\sum_{t=0}^{T-1}\mathbb{E}\left[\left(\frac{4D_1}{N_3} + \frac{D_1}{8N_3} + \frac{33q\epsilon^2 L_1^2}{N_4 L_0^2} + \frac{q}{8N_4} + \frac{qD_1}{2N_2 N_3} + \frac{4q^2\epsilon^2 L_1^2}{N_2 N_4 L_0^2}\right)(g_t)^2\right]$$

$$+ \left(\frac{1}{8L_2} + \frac{L_1}{50}\right)\sum_{t=0}^{T-1}\mathbb{E}\left[\left(\frac{D_2}{N_1} + \frac{2qL_2^2}{N_2 L_0^2}\epsilon^2 + \frac{q}{8N_2}(g_t)^2\right)\right]. \tag{57}$$

Let $c_0 = \max(32L_2, 8L_0), c_1 = \left(4 + \frac{8L_1^2 D_2}{L_0^2} + \frac{32L_1^2 D_2}{N_1 L_0^2} + \frac{16L_1^2 L_2}{5D_1 L_0^3}\right)$ and $c_2 = \max\left(\frac{1}{8L_2} + \frac{L_1}{50}, 1\right)$. For $N_3 \geq \max\left(\frac{200D_1 L_2}{L_0}, \frac{3c_0(D_0 + 4D_1 D_2)}{2L_0\epsilon^2}\right)$, $N_4 \geq \max\left(\frac{5qL_2}{L_0}, \frac{6qc_1 c_0}{L_0}\right)$, $\epsilon^2 \leq \min\left(\frac{L_0^3 N_4}{1480L_1^2 L_2 q}, \frac{L_0^4}{42L_1 c_0}, 1\right)$, $N_2 \geq \max\left(\frac{20qD_1 L_2}{L_0}, 20qc_2 L_2, \frac{12qL_2^2 c_0 c_2}{L_0^2}, q\right)$ and $N_1 \geq \frac{6D_2 c_0 c_2}{\epsilon^2}$, we then have that

$$\sum_{t=0}^{T-1}\mathbb{E}\left[\frac{1}{32L_2}(g_t)^2 + \frac{\beta_t}{4}\|v_t\|^2\right]$$

$$\leq \mathbb{E}[\mathcal{L}(x_0, \eta_0) - \mathcal{L}(x_T, \eta_T)] + \frac{\epsilon^2 T}{6c_0} + \frac{D_0 + 4D_1 D_2}{4L_0 N_3}T$$

$$+ \epsilon^2 T\frac{q}{N_4 L_0}\left(4 + \frac{8L_1^2 D_2}{L_0^2} + \frac{32L_1^2 D_2}{N_1 L_0^2} + \frac{16L_1^2 L_2}{5D_1 L_0^3}\right)$$

$$+ \left(\frac{1}{8L_2} + \frac{L_1}{50}\right)\frac{D_2 T}{N_1} + \left(\frac{1}{8L_2} + \frac{L_1}{50}\right)\epsilon^2 T\frac{2qL_2^2}{N_2 L_0^2}$$

$$\leq \mathbb{E}[\mathcal{L}(x_0, \eta_0) - \mathcal{L}(x_T, \eta_T)] + \frac{5\epsilon^2 T}{6c_0}. \tag{58}$$

For $T = nq \geq \frac{6c_0(\mathcal{L}(x_0, \eta_0) - \inf_{x,\eta}\mathcal{L}(x,\eta))}{\epsilon^2}$, we can find some $t_0 \mod q = 0$ such that

$$\sum_{t'=t_0}^{t_0+q-1}\mathbb{E}\left[\frac{1}{32L_2}(g_{t'})^2 + \frac{\beta_{t'}}{4}\|v_{t'}\|^2\right] \leq \frac{q\epsilon^2}{32L_2}. \tag{59}$$

Moreover we can find some $t \in [t_0, t_0 + q - 1)$ such that

$$\frac{1}{32L_2}\mathbb{E}[(g_t)^2] + \mathbb{E}[\frac{\beta_t}{4}\|v_t\|^2] \leq \frac{\epsilon^2}{c_0}. \tag{60}$$

Based on equation 60 and $c_0 = \max(32L_2, 8L_0)$, we have that $\mathbb{E}[(g_t)^2] \leq \epsilon^2$. Based on equation 67 and equation 68, we can further show that

$$\mathbb{E}[(g_t - \nabla_\eta \mathcal{L}(x_t, \eta_t))^2] \leq \frac{D_2}{N_1} + \sum_{t'=t_0}^{t_0+q-1} \mathbb{E}\left[\frac{2L_2^2}{N_2 L_0^2}\epsilon^2 + \frac{2L_2^2}{N_2}\alpha_{t-1}^2(g_{t'})^2\right]$$

$$\leq \frac{\epsilon^2}{6c_0} + \frac{\epsilon^2}{6c_0} + \frac{\epsilon^2}{160L_2} \leq \frac{\epsilon^2}{60L_2}. \tag{61}$$

Thus we have that

$$\mathbb{E}[|\nabla_\eta \mathcal{L}(x_t, \eta_{t+1})|] \leq \mathbb{E}[|g_t - \nabla_\eta \mathcal{L}(x_t, \eta_t)| + |g_t| + L_2\alpha_t|g_t|] \leq \left(\frac{5}{4} + \sqrt{\frac{1}{60L_2}}\right)\epsilon.$$

Moreover, we can show that

$$\frac{\epsilon^2}{L_0} \geq \mathbb{E}[\beta_t\|v_t\|^2] = \mathbb{E}\left[\frac{\epsilon^2}{L_0}\min\left(\frac{\|v_t\|^2}{2\epsilon^2}, \frac{\|v_t\|}{\epsilon}\right)\right] \geq \mathbb{E}\left[\frac{\epsilon\|v_t\|}{L_0}\right] - \frac{\epsilon^2}{2L_0}, \tag{62}$$

where the second inequality is due to the fact that $\min\left(\frac{a^2}{2}, a\right) \geq a - \frac{1}{2}$ holds for any $a \geq 0$. As a result, we have that $\mathbb{E}[\|v_t\|] \leq 3\epsilon$. Based on equation 71 and equation 72, we can further show that

$$\mathbb{E}[\|v_t - \nabla_x \mathcal{L}(x_t, \eta_{t+1})\|^2]$$

$$\leq \mathbb{E}\left[\frac{D_0 + 4D_1(g_{t_0})^2 + 2D_1 L_2^2 \alpha_{t_0}^2(g_{t_0})^2}{N_3} + \frac{4D_1(g_{t_0} - \nabla_\eta \mathcal{L}(x_{t_0}, \eta_{t_0}))^2}{N_3}\right]$$

$$+ \sum_{t'=t_0+1}^{t_0+q-1} \mathbb{E}\left[\frac{2}{N_4}\epsilon^2\left(2 + 8\frac{L_1^2 L_2^2}{L_0^2}\alpha_{t'-1}^2(g_{t'-1})^2 + 4\frac{L_1^2}{L_0^2}D_2\right) + \frac{2}{N_4}L_2^2\alpha_t^2(g_{t'})^2\right]$$

$$+ \frac{2\epsilon^2}{N_4}\sum_{t'=t_0+1}^{t_0+q-1} \mathbb{E}\left[16\frac{L_1^2}{L_0^2}(g_{t'-1})^2 + 16\frac{L_1^2}{L_0^2}(\nabla_\eta \mathcal{L}(x_{t'-1}, \eta_{t'-1}) - g_{t'-1})^2\right]$$

$$\leq \frac{D_0}{N_3} + \left(\frac{4D_1 + 2D_1 L_2^2 \alpha_{t_0}^2}{N_3} + \frac{\epsilon^2 L_1^2}{N_4 L_0^2} + \frac{L_2^2}{8N_4 L_0^2} + \frac{32\epsilon^2 L_1^2}{N_4 L_0^2}\right)\sum_{t'=t_0+1}^{t_0+q-1} \mathbb{E}[(g_{t'})^2]$$

$$+ \frac{4q\epsilon^2}{N_4} + \frac{8q\epsilon^2 L_1^2 D_2}{N_4} + \left(\frac{4D_1}{N_3} + \frac{32\epsilon^2 L_1^2}{N_4 L_0^2}\right)\sum_{t'=t_0}^{t_0+q-1} \mathbb{E}[(\nabla_\eta \mathcal{L}(x_{t'}, \eta_{t'}) - g_{t'})^2]$$

$$\leq \epsilon^2 + \left(\frac{D_1 L_0}{L_2} + \frac{33L_1^2}{5L_0 L_2} + \frac{L_2}{40L_0}\right)\epsilon^2 + \frac{4L_0}{5L_2}\epsilon^2 + \frac{8L_0 L_2 D_2}{5L_2}\epsilon^2 + \left(\frac{4D_1 L_0}{5L_2} + \frac{32L_1^2}{5L_0 L_2}\right)\frac{\epsilon^2}{15L_2}$$

$$\leq \left(1 + \frac{L_2}{40L_0} + \frac{L_0 D_1 + L_0 + 2L_0 L_2 D_2}{L_2} + \frac{33L_1^2}{5L_0 L_2} + \frac{L_0 D_1}{15L_2^2} + \frac{L_1^2}{2L_0 L_2^2}\right)\epsilon^2$$

$$\leq c_3 \epsilon^2. \tag{63}$$

Thus we have that

$$\mathbb{E}[\|\nabla_x \mathcal{L}(x_t, \eta_{t+1})\|] \leq \mathbb{E}[\|v_t - \nabla_x \mathcal{L}(x_t, \eta_{t+1})\| + \|v_t\|] \leq (3 + \sqrt{c_3})\epsilon.$$

It then follows that $\mathbb{E}[\|\nabla_{x,\eta} \mathcal{L}(x_t, \eta_{t+1})\|] \leq \mathbb{E}[|\nabla_\eta \mathcal{L}(x_t, \eta_{t+1})|] + \mathbb{E}[\|\nabla_x \mathcal{L}(x_t, \eta_{t+1})\|] \leq c_4\epsilon$. This completes the proof. □

### A.9 FULL VERSION AND ITS PROOF OF LEMMA 5

**Lemma 6.** *With the parameters selected in Theorem 4, for each $t_0 < T$ that can be divided by $q$, we have that*

$$\sum_{t=t_0}^{t_0+q-1} \mathbb{E}[(g_t - \nabla_\eta \mathcal{L}(x_t, \eta_t))^2] \leq \sum_{t=t_0}^{t_0+q-1} \left( \frac{D_2}{N_1} + \frac{2qL_2^2}{N_2 L_0^2}\epsilon^2 + \frac{2qL_2^2}{N_2}\alpha_t(g_t)^2 \right) \quad (64)$$

*and*

$$\sum_{t=t_0}^{t_0+q-1} \mathbb{E}[\|v_t - \nabla_x \mathcal{L}(x_t, \eta_{t+1})\|^2]$$

$$\leq \sum_{t=t_0}^{t_0+q-1} \mathbb{E}\left[ \frac{D_0}{N_3} + \frac{4D_1 D_2}{N_3 N_1} + \frac{q\epsilon^2}{N_4}\left( 4 + \frac{8L_1^2 D_2}{L_0^2} + \frac{32L_1^2 D_2}{N_1 L_0^2} + \frac{64q\epsilon^2 L_1^2 L_2^2}{N_2 L_0^4} \right) \right]$$

$$+ \sum_{t=t_0}^{t_0+q-1} \mathbb{E}\left[ \left( \frac{4D_1}{N_3} + \frac{D_1}{8N_3} + \frac{33q\epsilon^2 L_1^2}{N_4 L_0^2} + \frac{q}{8N_4} + \frac{qD_1}{2N_2 N_3} + \frac{4q^2\epsilon^2 L_1^2}{N_2 N_4 L_0^2} \right)(g_t)^2 \right]. \quad (65)$$

*Proof.* If $t \mod q = 0$, $g_t$ is an unbiased estimate of $\nabla_\eta \mathcal{L}(x_t, \eta_t)$ and according to Lemma 3, we have that

$$\mathbb{E}[(g_t - \nabla_\eta \mathcal{L}(x_t, \eta_t))^2] \leq \frac{D_2}{N_1}. \quad (66)$$

Otherwise, we have that

$$\mathbb{E}[(g_t - \nabla_\eta \mathcal{L}(x_t, \eta_t))^2]$$
$$= \mathbb{E}[(\nabla_\eta \mathcal{L}(x_t, \eta_t, \mathcal{B}_2) - \nabla_\eta \mathcal{L}(x_{t-1}, \eta_{t-1}, \mathcal{B}_2) + g_{t-1} - \nabla_\eta \mathcal{L}(x_t, \eta_t))^2]$$
$$= \mathbb{E}[(\nabla_\eta \mathcal{L}(x_t, \eta_t, \mathcal{B}_2) - \nabla_\eta \mathcal{L}(x_{t-1}, \eta_{t-1}, \mathcal{B}_2) + \nabla_\eta \mathcal{L}(x_{t-1}, \eta_{t-1})$$
$$- \nabla_\eta \mathcal{L}(x_t, \eta_t))^2] + \mathbb{E}[(g_{t-1} - \nabla_\eta \mathcal{L}(x_{t-1}, \eta_{t-1}))^2], \quad (67)$$

where the last inequality is due to the fact that $\nabla_\eta \mathcal{L}(x_t, \eta_t, \mathcal{B}_2) - \nabla_\eta \mathcal{L}(x_{t-1}, \eta_{t-1}, \mathcal{B}_2)$ is an unbiased estimate of $\nabla_\eta \mathcal{L}(x_t, \eta_t) - \nabla_\eta \mathcal{L}(x_{t-1}, \eta_{t-1})$. We now focus on the first term of equation 67, which can be further bounded as follows:

$$\mathbb{E}[(\nabla_\eta \mathcal{L}(x_t, \eta_t, \mathcal{B}_2) - \nabla_\eta \mathcal{L}(x_{t-1}, \eta_{t-1}, \mathcal{B}_2) + \nabla_\eta \mathcal{L}(x_{t-1}, \eta_{t-1}) - \nabla_\eta \mathcal{L}(x_t, \eta_t))^2]$$

$$\leq \frac{1}{N_2}\mathbb{E}[(\nabla_\eta \mathcal{L}(x_t, \eta_t, S) - \nabla_\eta \mathcal{L}(x_{t-1}, \eta_{t-1}, S))^2]$$

$$\leq \frac{2}{N_2}\mathbb{E}[(\nabla_\eta \mathcal{L}(x_t, \eta_t, S) - \nabla_\eta \mathcal{L}(x_{t-1}, \eta_t, S))^2 + (\nabla_\eta \mathcal{L}(x_{t-1}, \eta_t, S) - \nabla_\eta \mathcal{L}(x_{t-1}, \eta_{t-1}, S))^2]$$

$$\leq \mathbb{E}\left[ \frac{2}{N_2}L_2^2(\alpha_{t-1}^2(g_{t-1})^2 + \beta_{t-1}^2\|v_{t-1}\|^2) \right]$$

$$\leq \mathbb{E}\left[ \frac{2L_2^2}{N_2 L_0^2}\epsilon^2 + \frac{2L_2^2}{N_2}\alpha_{t-1}^2(g_{t-1})^2 \right], \quad (68)$$

where the first inequality is due to the fact that the square of expectation is not larger than the expectation of square, the third inequality is due to the continuous properties shown in Lemma 4, and the last inequality is due to the fact that $\beta_t = \min\left( \frac{1}{2L_0}, \frac{\epsilon}{L_0\|v_t\|} \right)$. Combining equation 66, equation 67, and equation 68, for $t_0 \mod q = 0$, we have that

$$\sum_{t=t_0}^{t_0+q-1} \mathbb{E}[(g_t - \nabla_\eta \mathcal{L}(x_t, \eta_t))^2] \leq \sum_{t=t_0}^{t_0+q-1} \mathbb{E}\left[ \left( \frac{D_2}{N_1} + \frac{2qL_2^2}{N_2 L_0^2}\epsilon^2 + \frac{q}{8N_2}(g_t)^2 \right) \right]. \quad (69)$$

We then focus on the estimate of the gradient to $x$. If $t \mod q = 0$, we have that

$$\mathbb{E}[\|v_t - \nabla_x \mathcal{L}(x_t, \eta_{t+1})\|^2]$$

$$\leq \mathbb{E}\left[ \frac{D_0 + D_1(\nabla_\eta \mathcal{L}(x_t, \eta_{t+1}))^2}{N_3} \right]$$

$$\leq \mathbb{E}\left[\frac{D_0 + 2D_1(\nabla_\eta \mathcal{L}(x_t, \eta_t))^2 + 2D_1 L_2^2 \alpha_t^2 (g_t)^2}{N_3}\right]$$

$$\leq \mathbb{E}\left[\frac{D_0 + 4D_1(g_t)^2 + 2D_1 L_2^2 \alpha_t^2 (g_t)^2}{N_3} + \frac{4D_1(g_t - \nabla_\eta \mathcal{L}(x_t, \eta_t))^2}{N_3}\right], \tag{70}$$

where the second inequality is due to the $L_2$-smoothness on $\eta$ and the update of $\eta$. Otherwise, we have that

$$\mathbb{E}[\|v_t - \nabla_x \mathcal{L}(x_t, \eta_{t+1})\|^2]$$

$$= \mathbb{E}[\|\nabla_x \mathcal{L}(x_t, \eta_{t+1}, \mathcal{B}_4) - \nabla_x \mathcal{L}(x_{t-1}, \eta_t, \mathcal{B}_4) + v_{t-1} - \nabla_x \mathcal{L}(x_t, \eta_{t+1})\|^2]$$

$$= \mathbb{E}[\|\nabla_x \mathcal{L}(x_t, \eta_{t+1}, \mathcal{B}_4) - \nabla_x \mathcal{L}(x_{t-1}, \eta_t, \mathcal{B}_4) + \nabla_x \mathcal{L}(x_{t-1}, \eta_t) - \nabla_x \mathcal{L}(x_t, \eta_{t+1})\|^2]$$

$$\quad + \mathbb{E}[\|v_{t-1} - \nabla_x \mathcal{L}(x_{t-1}, \eta_t)\|^2], \tag{71}$$

since $\nabla_x \mathcal{L}(x_t, \eta_{t+1}, \mathcal{B}_4) - \nabla_x \mathcal{L}(x_{t-1}, \eta_t, \mathcal{B}_4)$ is an unbiased estimate of $\nabla_x \mathcal{L}(x_t, \eta_{t+1}) - \nabla_x \mathcal{L}(x_{t-1}, \eta_t)$. We now focus on the first term of equation 71, which can be further bounded as follows:

$$\mathbb{E}[\|\nabla_x \mathcal{L}(x_t, \eta_{t+1}, \mathcal{B}_4) - \nabla_x \mathcal{L}(x_{t-1}, \eta_t, \mathcal{B}_4) + \nabla_x \mathcal{L}(x_{t-1}, \eta_t) - \nabla_x \mathcal{L}(x_t, \eta_{t+1})\|^2]$$

$$\leq \frac{1}{N_4} \mathbb{E}[\|\nabla_x \mathcal{L}(x_t, \eta_{t+1}, S) - \nabla_x \mathcal{L}(x_{t-1}, \eta_t, S)\|^2]$$

$$\leq \frac{2}{N_4} \mathbb{E}[\|\nabla_x \mathcal{L}(x_t, \eta_{t+1}, S) - \nabla_x \mathcal{L}(x_t, \eta_t, S)\|^2 + \|\nabla_x \mathcal{L}(x_t, \eta_t, S) - \nabla_x \mathcal{L}(x_{t-1}, \eta_t, S)\|^2]$$

$$\leq \mathbb{E}\left[\frac{2}{N_4} \mathbb{E}[(L_2^2 \alpha_t^2 (g_t)^2 + (2L_0^2 + 2L_1^2|\nabla_\eta \mathcal{L}(x_{t-1}, \eta_t, S)|^2)\beta_{t-1}^2 \|v_{t-1}\|^2)]\right]$$

$$\leq \mathbb{E}\left[\frac{2}{N_4} \epsilon^2 \left(2 + 4\frac{L_1^2}{L_0^2}(\nabla_\eta \mathcal{L}(x_{t-1}, \eta_t))^2 + 4\frac{L_1^2}{L_0^2}D_2\right) + \frac{2}{N_4}L_2^2 \alpha_t^2 (g_t)^2\right]$$

$$\leq \mathbb{E}\left[\frac{2}{N_4} \epsilon^2 \left(2 + 8\frac{L_1^2}{L_0^2}(\nabla_\eta \mathcal{L}(x_{t-1}, \eta_{t-1}))^2 + 8\frac{L_1^2 L_2^2}{L_0^2}\alpha_{t-1}^2 (g_{t-1})^2 + 4\frac{L_1^2}{L_0^2}D_2\right) + \frac{2}{N_4}L_2^2 \alpha_t^2 (g_t)^2\right]$$

$$\leq \mathbb{E}\left[\frac{2}{N_4} \epsilon^2 \left(2 + 8\frac{L_1^2 L_2^2}{L_0^2}\alpha_{t-1}^2 (g_{t-1})^2 + 4\frac{L_1^2}{L_0^2}D_2\right) + \frac{2}{N_4}L_2^2 \alpha_t^2 (g_t)^2\right]$$

$$\quad + \frac{2\epsilon^2}{N_4} \mathbb{E}\left[16\frac{L_1^2}{L_0^2}(g_{t-1})^2 + 16\frac{L_1^2}{L_0^2}(\nabla_\eta \mathcal{L}(x_{t-1}, \eta_{t-1}) - g_{t-1})^2\right], \tag{72}$$

where the third inequality is due to Lemma 4 , the fourth inequality is due to $\beta_t = \min\left(\frac{1}{2L_0}, \frac{\epsilon}{L_0\|v_t\|}\right)$ and Lemma 3. Combine equation 70, equation 71 and equation 72, and for $t_0 \mod q = 0$, we have that

$$\sum_{t=t_0}^{t_0+q-1} \mathbb{E}[\|v_t - \nabla_x \mathcal{L}(x_t, \eta_{t+1})\|^2]$$

$$\leq \sum_{t=t_0}^{t_0+q-1} \mathbb{E}\left[\frac{D_0}{N_3} + \frac{q\epsilon^2}{N_4}\left(4 + \frac{8L_1^2 D_2}{L_0^2}\right)\right]$$

$$\quad + \sum_{t=t_0}^{t_0+q-1} \mathbb{E}\left[\left(\frac{4D_1}{N_3} + \frac{D_1}{8N_3} + \frac{33q\epsilon^2 L_1^2}{N_4 L_0^2} + \frac{q}{8N_4}\right)(g_t)^2\right]$$

$$\quad + \sum_{t=t_0}^{t_0+q-1} \mathbb{E}\left[\left(\frac{4D_1}{N_3} + \frac{32q\epsilon^2 L_1^2}{N_4 L_0^2}\right)(g_t - \nabla_\eta \mathcal{L}(x_t, \eta_t))^2\right]$$

$$\leq \sum_{t=t_0}^{t_0+q-1} \mathbb{E}\left[\frac{D_0}{N_3} + \frac{q\epsilon^2}{N_4}\left(4 + \frac{8L_1^2 D_2}{L_0^2}\right)\right]$$

$$\quad + \sum_{t=t_0}^{t_0+q-1} \mathbb{E}\left[\left(\frac{4D_1}{N_3} + \frac{D_1}{8N_3} + \frac{33q\epsilon^2 L_1^2}{N_4 L_0^2} + \frac{q}{8N_4}\right)(g_t)^2\right]$$

$$+ \left( \frac{4D_1}{N_3} + \frac{32q\epsilon^2 L_1^2}{N_4 L_0^2} \right) \sum_{t=t_0}^{t_0+q-1} \mathbb{E}\left[ \left( \frac{D_2}{N_1} + \frac{2qL_2^2}{N_2 L_0^2}\epsilon^2 + \frac{2qL_2^2}{N_2}\alpha_t^2(g_t)^2 \right) \right]$$

$$\leq \sum_{t=t_0}^{t_0+q-1} \mathbb{E}\left[ \frac{D_0}{N_3} + \frac{4D_1 D_2}{N_3 N_1} + \frac{q\epsilon^2}{N_4}\left( 4 + \frac{8L_1^2 D_2}{L_0^2} + \frac{32L_1^2 D_2}{N_1 L_0^2} + \frac{64q\epsilon^2 L_1^2 L_2^2}{N_2 L_0^4} \right) \right]$$

$$+ \sum_{t=t_0}^{t_0+q-1} \mathbb{E}\left[ \left( \frac{4D_1}{N_3} + \frac{D_1}{8N_3} + \frac{33q\epsilon^2 L_1^2}{N_4 L_0^2} + \frac{q}{8N_4} + \frac{qD_1}{2N_2 N_3} + \frac{4q^2\epsilon^2 L_1^2}{N_2 N_4 L_0^2} \right) (g_t)^2 \right]. \qquad (73)$$

This completes the proof. $\qquad\square$

## A.10    DETAILS OF THE EXPERIMENT

### A.10.1    LIFE EXPECTANCY DATA

This life expectancy data includes health factors from 193 countries (input features) and the life expectancy (target) from the World Health Organization and United Nations websites. We follow the same data pre-processing as in Chen et al. (2023) to fill the missing data with the medians, censorize and standardize all the features [2], remove redundant features, and a standard Gaussian noise is added to the target for model robustness. Each element of the initial parameter $x_0$ is generated by a standard Gaussian distribution. In our deterministic setting, we compare the methods with fine-tuned learning rates. The iteration number is set to 50. For existing methods, we follow the fine-tuned learning rates in (Chen et al., 2023), where the step size $\beta_t = 10^{-4}$ for GD, $\beta_t = 0.2$ for normalized GD and $\beta_t = 0.3\min\left( \frac{1}{10}, \frac{1}{\|\nabla_{x,\eta}\mathcal{L}(x_t, \eta_t)\|} \right)$. For our D-GD method, we set $\alpha_t = \beta_t = 10^{-4}$ and for our D-GD-C method, we set $\alpha_t = 10^{-4}$ and $\beta_t = 0.35\min\left( \frac{1}{2000}, \frac{1}{\|\nabla_x \mathcal{L}(x_t, \eta_{t+1})\|} \right)$.

In our stochastic setting, we run the experiments for 5000 iterations. We generate the initial $x_0, \eta_0$ by running a normalized GD method with step size $\beta_t = 0.2$ for 30 iterations. For existing methods, we follow the same setting in (Chen et al., 2023). We set the mini-batch size to 50. For SGD, the step size is $\beta_t = 2 \times 10^{-4}$. For the normalized SGD with momentum method, the momentum coefficient is set to $10^{-4}$ and the step size is set to $8 \times 10^{-3}$. For the normalized SPIDER method, we have that step size $\beta_t = 4 \times 10^{-3}$ and epoch size $q = 20$. For our D-SGD-C, we set $\alpha_t = 8 \times 10^{-5}$ and $\beta_t = 0.05\min(\frac{1}{100}, \frac{1}{\|v_t\|})$. For our D-SPIDER-C, we have that $\alpha_t = 8 \times 10^{-5}$ and $\beta_t = 7.5 \times 10^{-3}\min\left( 2.5, \frac{1}{\|v_t\|} \right)$.

### A.10.2    CIFFAR-10 DATASET

In this part, we conduct experiments on the famous CIFAR-10 dataset (Alex, 2009), which includes 50000 training samples. We employ DRO model to construct a linear classifier. After extracting features from a pre-trained ResNet-50 model (He et al., 2016), for the $i$-th sample in our dataset, we have input $z_i \in \mathbb{R}^{2049}$ and output $y_i \in [10]$. The non-convex original loss function is set as $\ell(x, (z_i, y_i)) = \ln(\sum_{j=1}^{10} \exp(x_j^\top z_i - x_{y_i}^\top z_i)) + 0.001\sum_{j=1}^{10}\sum_{k=1}^{2049}\ln(1 + |x_{j,k}|)$, where $x \in \mathbb{R}^{10 \times 2049}$ is the trainable parameter, and $x_j$ is the $j$-th row. For the DRO model, $\lambda$ is set to $0.05$, and the initial value $\eta_0$ is set to $0.1$. Each element of the initial parameter $x_0$ is generated by a standard Gaussian distribution.

In Figure 3, we provide the training curves with fine-tuned learning rate for SGD, Normalized-SPIDER (Chen et al., 2023), Normalized-SGD with momentum (Jin et al., 2021) and our proposed D-SGD-C and D-SPIDER-C methods. The x-axis stands for the training iteration, and y-axis stands for the DRO dual function value. For Figure 3 we can observe that the Normalized-SPIDER and our D-SPIDER-C have similar performance and converge faster than other methods. Our D-SGD-C has a similar performance compared with the Normalized-SGD with momentum method but our D-SPIDER-C outperforms the Normalized-SGD with momentum method.

We set the mini-batch size to 200. For SGD, the step size is $\beta_t = 1.5 \times 10^{-3}$. For the normalized SGD with momentum method, the momentum coefficient is set to $0.1$ and the step size is set to

---

[2]https://thecleverprogrammer.com/2021/01/06/life-expectancy-analysis-with-python/

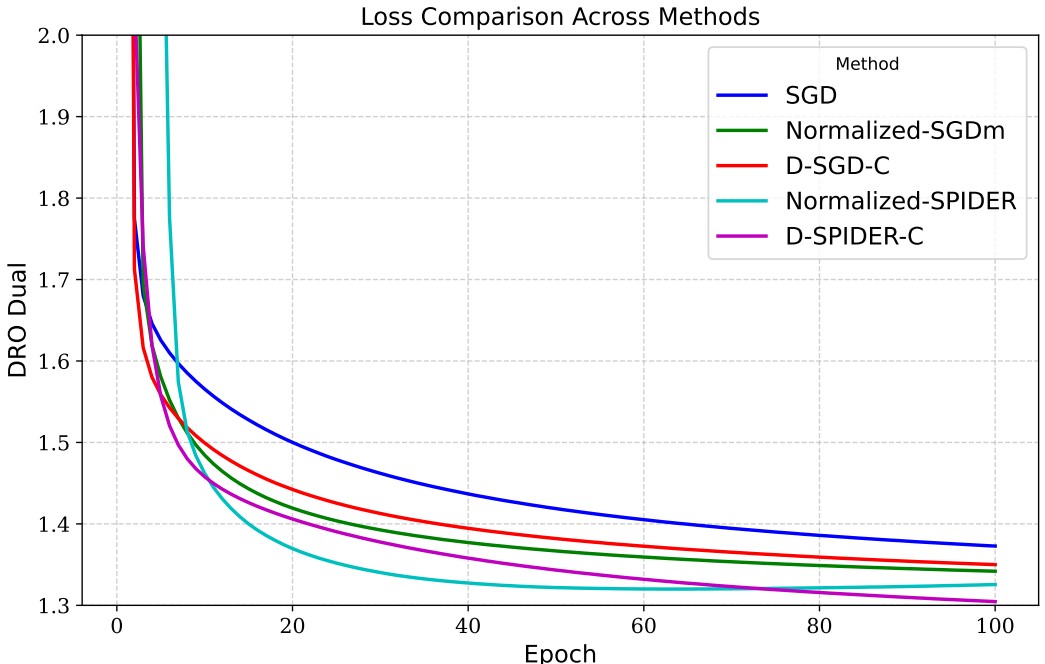

Figure 3: The numerical results for Cifar-10 dataset.

$4 \times 10^{-3}$. For the normalized SPIDER method, we have that step size $\beta_t = 3.5 \times 10^{-3}$ and epoch size $q = 10$ and large batch size $2 \times 10^3$. For our D-SGD-C, we set $\alpha_t = 1 \times 10^{-3}$ and $\beta_t = \min(\frac{2}{1000}, \frac{1}{10\|v_t\|})$. For our D-SPIDER-C, we have that $\alpha_t = 1 \times 10^{-3}$ and $\beta_t = \min\left(\frac{5}{2000}, \frac{1}{20\|v_t\|}\right)$.

### A.11 RESULTS FOR NORMALIZED MOMENTUM

---
**Algorithm 4** D-SGD-M
---
**Input:** initialization $x_0, \eta_0$, step sizes $\alpha, \beta, r_1, r_2$ number of interactions $T$

1: $t \leftarrow 1$
2: **while** $t \leq T$ **do**
3:     Draw one sample $s$ from $P_0$ and compute $g_{t-1} \leftarrow \nabla_\eta \mathcal{L}(x_{t-1}, \eta_{t-1}, s)$
4:     **if** t==1 **then**
5:         Draw one sample $s$ from $P_0$ and $m_0 \leftarrow \nabla_\eta \mathcal{L}(x_{t-1}, \eta_{t-1}, s)$
6:     **end if**
7:     $m_t \leftarrow r_1 m_{t-1} + (1 - r_1)g_{t-1}$
8:     $\eta_t \leftarrow \eta_{t-1} - \alpha \frac{m_t}{\|m_t\|}$
9:     Draw one sample $s$ from $P_0$ and compute $v_{t-1} \leftarrow \nabla_x \mathcal{L}(x_{t-1}, \eta_t, s)$
10:     **if** t==1 **then**
11:         Draw one sample $s$ from $P_0$ and $w_0 \leftarrow \nabla_x \mathcal{L}(x_{t-1}, \eta_t, s)$
12:     **end if**
13:     $w_t \leftarrow r_2 w_{t-1} + (1 - r_2)v_{t-1}$
14:     $x_t \leftarrow x_{t-1} - \beta \frac{w_t}{\|w_t\|}$
15:     $t \leftarrow t + 1$
16: **end while**
---

In this section, we provide result for the normalized momentum method, which is shown in Algorithm 4. The following proposition establishes the convergence and complexity of Algorithm 4.

**Proposition 1.** *Set $1 - r_1, 1 - r_2 \leq \mathcal{O}(\epsilon^2)$, $\beta \leq \mathcal{O}(\epsilon^{-3}), \alpha = \mathcal{O}(\sqrt{D_1}\beta)$ and $T \geq \mathcal{O}(\epsilon^{-4})$ for Algorithm 4. We then have that*

$$\min_{t<T} \mathbb{E}[\|\nabla_{x,\eta}\mathcal{L}(x_t, \eta_{t+1})\|] \leq \mathcal{O}(\epsilon).$$

Note that compared with the normalized momentum algorithm for generalized smooth objective (Jin et al., 2021), Algorithm 4 does not require the full gradient information for initialization or the use of mini-batches. This is because that due to our partially generalized smoothness, the expectations of $\|m_0 - \mathcal{L}(x_0, \eta_0)\|$ and $\|v_0 - \mathcal{L}(x_0, \eta_1)\|$ are no longer unbounded thus the requirement on full gradient is not needed. Moreover, the partially generalized smoothness allows the Lipschitz constant on $x$ bounded by a linear function of gradient on $\eta$. Thus by setting $\alpha = \mathcal{O}(\sqrt{D_1}\beta)$ we can remove the mini-batch required in Jin et al. (2021).

