# OpenReview forum: "Revisiting Large-Scale Non-convex Distributionally Robust Optimization"
_ICLR.cc/2025/Conference — ICLR 2025 Poster_

### Official Review · Reviewer_khoB · 2024-11-01

**Soundness:** 3
**Presentation:** 2
**Contribution:** 3
**Rating:** 5
**Confidence:** 3

**Summary:**

This paper studies Distributionally Robust Optimization (DRO) with non-convex smooth functions, which optimizes the expected loss under the worst-case distribution in an uncertainty set of distributions, which is different from the tradition Empirical Risk Minimization (ERM). In particular, the paper improves Jin et al (2021) which treats the same problem, working with the dual formulation. The authors show that the dual problem and the gradient noise satisfy simpler yet more precise partially generalized smoothness condition and partially affine variance condition by studying the optimization variable and dual variable separately. This also yields simpler algorithm design and convergence analysis, based on which the authors propose D-SGD-C algorithm and variance reduced SGD versions.

**Strengths:**

(1) Proposed an improvement of Jin et al (2021) and showed that by studying the optimization variable and the dual variable separately, one can derive the fact that the dual problem and the gradient noise satisfy simpler yet more precise partially generalized smoothness condition and partially affine variance condition;

(2) Based on this idea, they also propose new and simpler algorithms like D-SGD-C etc.

**Weaknesses:**

(1) This is an almost pure math paper, so a better explanation of the main mathematical ideas at the introductory part would be nice

(2) no experimental validations

**Questions:**

After reading the paper, I feel that the main context is a dry list of theorems referring to the proofs in the appendix. So the mathematical innovations can be lost if there is no better overview of the ideas. Can the authors provide one in a revision?

---

> ### Author Response · Authors · 2024-11-21
>
> We thank the reviewer khoB for the valuable feedback and here are our responses.
>
>  **Weakness 1:**  Thanks for pointing it out.
>
> * We have followed your suggestion and provided an overview of the ideas in the introduction:
>
> Our proposed more precise conditions circumvent the unbounded Lipschitz constant and unbounded noise variance challenges in the generalized $(L_0,L_1)$-smoothness problems with affine variance noises. Our conditions first show the dual problem is standard $L$-smooth in the dual variable $\eta$ and the noise of gradient on $\eta$ has bounded variance. Then under our partially generalized smoothness and partially affine variance noise condition, the  Lipschitz constant and noise variance on the model parameter $x$ are also bounded, making the objective easy to solve.
>
>
> * In the main context, we have discussed in details the motivation of our approach, algorithm, key challenges and proof sketches. We have highlighted them in red.
>
>  **Weakness 2:**  Thanks for pointing it out.  In Section 4: numerical results, we provide the experiment on the life expectancy data. In our revision, we further added numerical results on large datasets (see Fig. 3). Based on the results, we can observe that the
> Normalized-SPIDER and our D-SPIDER-C have similar performance and converge faster than other methods.
> Our D-SGD-C has a similar performance compared with the Normalized-SGD with momentum method but our D-SPIDER-C outperforms the Normalized-SGD with momentum method.
>
>  **Question:**  Please see response to **weakness 1**.

---

> > ### Comment · Reviewer_khoB · 2024-11-21
> > **keep the score**
> >
> > Thank you very much! I keep the score as is.

---

> > > ### Author Response · Authors · 2024-11-22
> > >
> > > Dear Reviewer kohB,
> > >
> > > Thank you so much for your prompt response! We really appreciate the time you’ve spent reviewing our paper. If you have any other questions or concerns, please let us know—we’re happy to address them.
> > >
> > > If there aren’t any, we’d be very grateful if you could reconsider our score. Thanks again for your thoughtful feedback and support!
> > >
> > > Best,
> > > Authors

---

### Official Review · Reviewer_Rvfr · 2024-11-01

**Soundness:** 3
**Presentation:** 3
**Contribution:** 3
**Rating:** 6
**Confidence:** 3

**Summary:**

This paper studies the Distributionally robust optimization (DRO) problem. The paper first shows an observation that the specific DRO problem satisfies a stronger smoothness and noise condition than existing work showed. Then the paper designs multiple algorithms to solve the problem making use of the stronger assumptions, together with corresponding convergence guarantees.

**Strengths:**

1. The paper is well written, with a clear introduction to the studied problem and well-defined notations. I enjoy reading it.

2. The theory is neat and complete, taking 3 algorithms into consideration.

3. I appreciate the idea of finding that the problem actually satisfies stronger assumptions and designing approaches based on this observation.

**Weaknesses:**

1. The convergence in Theorem 3 is not strong enough, requiring a very large batch size. As a comparison, (Jin et. al.) achieved a convergence bound with a much more reasonable batch size requirement. In fact, I think this is partially why they used this more complex method, as momentum is proven to help handle noise in normalized gradient descent and enable convergence with small batch size (Cutkosky \& Mehta, 2020).

2. All the convergence results seem to require a large batch, and I don't see benefits compared to the results in (Jin et. al.). This makes me wonder whether the observation in Sections 3.1 and 3.3 that DRO satisfies stronger conditions really helpful for designing algorithms. I will be happy to raise my score if the authors can address my concerns.

[1] Jikai Jin, Bohang Zhang, Haiyang Wang, and Liwei Wang. Non-convex Distributionally Robust Optimization: Non-Asymptotic Analysis. In NeurIPS, 2021.

[2] Ashok Cutkosky and Harsh Mehta. Momentum Improves Normalized SGD. In ICML, 2020.

**Questions:**

1. It seems that the citation of the original spider paper (Fang et. al.) is missing. I kindly suggest the authors to add it.

2. Definition 3 actually indicates a much stronger version of the $(L_0,L_1)$-smoothness, as the inequality only applies to $||x-y|| \le 1/L_1$ for initial $(L_0,L_1)$-smoothness. Therefore, some expressions and citations in the paper regarding the assumption may be inappropriate (e.g. citations in line 192). I suggest the authors to add some discussions on this.

3. I think possibly momentum is also helpful for convergence with smaller batch sizes in your case. Have you tried this?

[1] Cong Fang, Chris Junchi Li, Zhouchen Lin, and T. Zhang, “SPIDER: Near-optimal non-convex optimization via stochastic path-integrated differential estimator.” In NeurIPS 2018.

---

> ### Author Response · Authors · 2024-11-21
>
> We thank the reviewer Rvfr for the valuable feedback and here are our responses.
>
> **Weakness 1:** Thanks for the question.
>
> * The convergence in [1] is not stronger than the convergence in this paper. In the last part of their proof for the main theorem in Section C.2.3,  the $m_0$ is required to be set to $\nabla F(x_0)$, which is the full gradient. No matter of the batch size, the full gradient is always required for the analysis in [1]. The momentum method is able to handle noise in normalized gradient descent and the sum of noise can be upper bounded by a function of $\|m_0-\nabla F(x_0)\|$. In the original normalized momentum paper [2], the gradient noise is assumed to have a bounded variance. Thus the expectation of $\|m_0-\nabla F(x_0)\|$ is bounded and there is no additional requirements on $m_0$. However, in [1] it is shown that for DRO problem, the dual formulation has an affine variance noise, which means the expectation of $\|m_0-\nabla F(x_0)\|$ is potentially unbounded. Thus even a large mini-batch can not bound the expectation of $\|m_0-\nabla F(x_0)\|$ and the authors in [1] uses the full gradient to get around this issue.
>
> * We agree with the reviewer that using momentum helps to achieve convergence with a moderate batch size. The momentum method can also be incorporated into the design of our algorithm. This can eliminate the need for mini-batches and requirements of full gradient for initialization with the same complexity.  We have revised our paper to reflect on the above discussion and remove the claim that we have a simpler algorithmic design. We add Algorithm 4 and Proposition 1 in Appendix A.11. We show that the same complexity can be achieved without using mini-batches.
>
> * We still include the algorithm and theorem for the minibatch method in the paper for the goodness of presenting the variance-reduction method. A large batch size is needed for the variance-reduction method, and in Theorem 4, we reduce the computational complexity from $\mathcal O(\epsilon^{-4})$ to $\mathcal O(\epsilon^{-3})$. Under our more precise and fundamental characterization of the dual of non-convex DRO, our proof is much simpler than [2] and result is stronger than [3].
>
> * Our method does not require the full gradient for the initial point, which is a significant advantage compared to [1] when the dataset is large.
>
> [1] Jikai Jin, Bohang Zhang, Haiyang Wang, and Liwei Wang. Non-convex distributionally robust
> optimization: Non-asymptotic analysis. In Proceedings of Advances in Neural Information Pro-
> cessing Systems, volume 34, pp. 2771–2782, 2021.
>
> [2] Ziyi Chen, Yi Zhou, Yingbin Liang, and Zhaosong Lu. Generalized-smooth nonconvex optimization
> is as efficient as smooth nonconvex optimization. arXiv preprint arXiv:2303.02854, 2023.
>
> [3] Amirhossein Reisizadeh, Haochuan Li, Subhro Das, and Ali Jadbabaie. Variance-reduced clipping
> for non-convex optimization. arXiv preprint arXiv:2303.00883, 2023.
>
> **Weakness 2:** Thanks for the question.
> As we discussed in response to **weakness 1**, we agree with the reviewer that using momentum helps to eliminate the need for large batch size. Therefore, we removed our claim in this paper that we provide a simpler algorithmic design. However, we would also like to highlight that our more precise and fundamental characterization of the dual problem: partially generalized smoothness and partially affine variance noise, enables much simpler and elegant convergence and complexity proofs. This is still of great importance in advancing the fundamental understanding of non-convex DRO.
>
> Moreover, some specific advantages of our approach over the one in [1] as listed below:
>
> * As we discussed in response to **Weakness 1**, our SGD method does not require the full gradient information for the initial point, but [1] requires the full gradient information. Based on our new understanding, for the normalized momentum method, the requirement of full gradient information for the initial point can be waived thanks to our partially affine noise variance condition after obtaining an $\eta$ with bounded $|\nabla_\eta \mathcal L|$. Moreover, the use of mini-batches is not required due to the bound on Lipschitz constant on $x$ does not depend on the gradient on $x$. See details in Prop 1 in the revision.
>
> * As we show in Theorem 4, the variance reduction method needs a large batch size and compared with the method in [1], our variance-reduced method reduces the computational complexity from $\mathcal O(\epsilon^{-4})$ to $\mathcal O(\epsilon^{-3})$.
>
> [1] Jikai Jin, Bohang Zhang, Haiyang Wang, and Liwei Wang. Non-convex distributionally robust
> optimization: Non-asymptotic analysis. In Proceedings of Advances in Neural Information Pro-
> cessing Systems, volume 34, pp. 2771–2782, 2021.

---

> > ### Author Response · Authors · 2024-11-21
> >
> > **Question 1:** Thanks for the reminder.  We have added it in the revision.
> >
> > **Question 2:** Thanks for the question.  We have added the following discussions in the revision.
> > Note that there are two versions of the $(L_0,L_1)$-smoothness, one requires the inequality holds for $\|x-y\|\le \frac{1}{L_0}$ ([1]) and one applies for all $x$ and $y$ ([2]). In DRO problems, it can be proved that for the dual objective ([3]), the inequality holds for any $x,y \in \mathbb R^d$, and therefore we follow the second definition in this paper.
> >
> > [1] Jingzhao Zhang, Tianxing He, Suvrit Sra, and Ali Jadbabaie. Why gradient clipping accelerates
> > training: A theoretical justification for adaptivity. arXiv preprint arXiv:1905.11881, 2019.
> >
> > [2] Ziyi Chen, Yi Zhou, Yingbin Liang, and Zhaosong Lu. Generalized-smooth nonconvex optimization is as efficient as smooth nonconvex optimization. arXiv preprint arXiv:2303.02854, 2023.
> >
> > [3] Jikai Jin, Bohang Zhang, Haiyang Wang, and Liwei Wang. Non-convex distributionally robust optimization: Non-asymptotic analysis. In Proceedings of Advances in Neural Information Pro- cessing Systems, volume 34, pp. 2771–2782, 2021.
> >
> > **Question 3:** Thanks for your insights. We have added Algorithm 4 and Proposition 1 in Appendix A.11 showing that using momentum, the same convergence and complexity can be achieved without using mini-batch.

---

> > > ### Comment · Reviewer_Rvfr · 2024-11-22
> > >
> > > Thanks for the update. I think currently the contribution and presentation is basically satisfying. And I appreciate the authors' effort in improving the work. Therefore, I have decided to raise my score to 6.

---

> > > > ### Author Response · Authors · 2024-11-22
> > > >
> > > > Dear Reviewer Rvfr,
> > > >
> > > > Thank you for your thoughtful insights and prompt response. We’re grateful for your feedback, which has helped us improve the work.
> > > >
> > > > Best regards,
> > > > The Authors

---

### Official Review · Reviewer_oGQa · 2024-11-04

**Soundness:** 3
**Presentation:** 3
**Contribution:** 2
**Rating:** 6
**Confidence:** 2

**Summary:**

This paper studies the penalized distributionally robust optimization (DRO) problem where the loss function $\ell(x, s)$ is $G$-Lipschitz and $L$-smooth in $x$ for any given sample $s$. In particular, the authors consider $\psi$-divergence distance in the penalty term and solve the DRO problem from its dual optimization. The authors refines the generalized smoothness used in prior work [1], and proposed alternating gradient methods (in terms of the dual variables and the primal variables) by noticing that the dual problem is actually smooth with respect to the dual variables. By this refined perspective of generalized smoothness, the authors achieve the same complexities as ones in [1] by simpler algorithms. Further convergence analysis under stochastic settings and variance reduction is conducted.

[1] Jikai Jin, Bohang Zhang, Haiyang Wang, and Liwei Wang. Non-convex distributionally robust optimization: Non-asymptotic analysis. In Proceedings of Advances in Neural Information Processing Systems, volume 34, pp. 2771–2782, 2021.

**Strengths:**

- The write-up of this paper is good and easy to follow.
- It is interesting to see that one can refine the smoothness on the dual problem of penalized DRO and develop simpler algorithms by exploiting the refined smoothness.
- The analysis is provided under both deterministic and stochastic settings. Improved complexity is achieved by recursive variance reduction.

**Weaknesses:**

- The generalized smoothness on the dual side depends on the strict assumptions on the primal loss function: assuming Lipschitzness and smoothness at the same time and for any sample $s \in S$; the bounded variance of the function values.
- The deterministic setting would be restricted, as one usually cannot access the full gradient of the stochastic objective like $\mathcal{L}$.
- Although the authors propose simpler algorithms, the step size requirements are actually strict: $\beta_t$ either dependents on the initial function gap (Theorem 1) or requires full gradient evaluation (Theorem 2).
- I would not say optimizing $x$ and $\eta$ separately; instead the algorithms are performing alternating gradient steps with respect to these two blocks of variables.

**Questions:**

- The paper title is quite vague. I suggest that the authors could rephrase it to make it more precise, like in terms of generalized smoothness.
- Could the authors discuss the technical challenges of this work, compared to the prior work [1]? It seems that the write-up and the analysis are mostly similar.
- It seems that generalized smoothness condition heavily relies on the Lipschitzness of the primal function. Am I understanding correctly, or it can be relaxed?
- In the stochastic settings, the step sizes $\beta_t = O(\varepsilon / L_0||v_t||)$. Could the authors discuss the dependence on $\varepsilon$?


[1] Jikai Jin, Bohang Zhang, Haiyang Wang, and Liwei Wang. Non-convex distributionally robust optimization: Non-asymptotic analysis. In Proceedings of Advances in Neural Information Processing Systems, volume 34, pp. 2771–2782, 2021.

---

> ### Author Response · Authors · 2024-11-21
>
> We thank the reviewer oGQa for the valuable feedback and here are our responses.
>
> **Weakness 1:**  Thanks for pointing it out. The reviewer is correct that to derive the partially generalized smoothness condition in this paper, we require the Lipschitzness and smoothness of the loss function. However, we would like to clarify that the assumptions in our paper are the **weakest** ones among all existing non-convex DRO studies. Below, we provide two tables that summarize the assumptions used in DRO. The first table is for DRO in the convex setting, and the second one is for DRO in the non-convex setting.
> | Method| Lipschitzness  |Boundedness  |
> |-|-|-|
> |[1]|required|required|
> |[2]|required|required|
> |[3]|required|required|
> |[4]|required|required|
>
> *Table: Assumptions used in convex DRO setting.*
>
> | Method| Lipschitzness  |Boundedness  |Smoothness|
> |-|-|-|-|
> |[5]|not required|not required|required|
> |[6]|required|required|required|
> |[7]|required|required|required|
> |[8]|required|required|required|
> |[9]|required|not required|required|
> |**This paper**|required|not required|required|
>
> *Table: Assumptions used in non-convex DRO setting.*
>
> Note that [5] does not require Lipschitz or bounded loss, but it requires $\lambda$ to be extremely large so that the dual function is strongly concave. The smoothness and Lipschitzness conditions are required in all other non-convex DRO studies. [9] and our paper require the mildest assumptions in the literature, where the loss function can be potentially unbounded and has a bounded variance.
>
> Reference:
>
> [1] Hongseok Namkoong and John C Duchi. Stochastic gradient methods for distributionally robust
> optimization with f-divergences. In Proceedings of Advances in neural information processing
> systems, volume 29, 2016.
>
> [2] John Duchi and Hongseok Namkoong. Learning models with uniform performance via distribution-
> ally robust optimization. arXiv preprint arXiv:1810.08750, 2018.
>
> [3] Daniel Levy, Yair Carmon, John C Duchi, and Aaron Sidford. Large-scale methods for distribution-
> ally robust optimization. In Proceedings of Advances in Neural Information Processing Systems,
> volume 33, pp. 8847–8860, 2020.
>
> [4] Jie Wang, Rui Gao, and Yao Xie. Sinkhorn distributionally robust optimization. arXiv preprint
> arXiv:2109.11926, 2021.
>
> [5] Aman Sinha, Hongseok Namkoong, Riccardo Volpi, and John Duchi. Certifying some distributional
> robustness with principled adversarial training. arXiv preprint arXiv:1710.10571, 2017.
>
> [6] Qi Qi, Zhishuai Guo, Yi Xu, Rong Jin, and Tianbao Yang. An online method for a class of distribu-
> tionally robust optimization with non-convex objectives. In Proceedings of Advances in Neural
> Information Processing Systems, volume 34, pp. 10067–10080, 2021.
>
> [7] Qi Qi, Jiameng Lyu, Er Wei Bai, Tianbao Yang, et al. Stochastic constrained dro with a complexity
> independent of sample size. arXiv preprint arXiv:2210.05740, 2022.
>
> [8] Qi Zhang, Yi Zhou, Ashley Prater-Bennette, Lixin Shen, and Shaofeng Zou. Large-scale non-convex
> stochastic constrained distributionally robust optimization. arXiv preprint arXiv:2404.01200,
> 2024a.
>
> [9] Jikai Jin, Bohang Zhang, Haiyang Wang, and Liwei Wang. Non-convex distributionally robust
> optimization: Non-asymptotic analysis. In Proceedings of Advances in Neural Information Pro-
> cessing Systems, volume 34, pp. 2771–2782, 2021.
>
> **Weakness 2:**
>
> We present the deterministic setting in order for a progressive introduction to our major technical developments. Results for the deterministic setting present algorithmic design and analysis in order to tackle the challenge of partially generalized smoothness without introducing further complications from partially affine variance noise in the stochastic setting. Results in the deterministic setting are also of practical importance when we have a relatively small dataset.
>
> We would also like to clarify that this paper presents results for the stochastic setting in Section 3.3, which addresses the limitations, e.g., limited access to full gradient, as mentioned by the reviewer.

---

> > ### Author Response · Authors · 2024-11-21
> >
> > **Weaknesses 3:**
> >
> > Thanks for pointing it out.
> >
> > The step size in Theorem 1 depends on $\mathcal L(x_0,\eta_0)-\inf_{x,\eta} \mathcal L(x,\eta)$, which is a constant and can be easily calculated in practice. Specifically, due to the strong duality, we have that $\inf_{x,\eta} \mathcal L(x,\eta)=\inf_x\sup_{Q}\mathbb E_{S\sim Q}[\ell(x,S)]-\lambda D(Q||P_0)\ge\inf_{x}  \mathbb E_{S\sim P_0}[\ell(x,S)]$. Thus for a non-negative loss $\ell$, we have $\inf_{x,\eta} \mathcal L(x,\eta)$ non-negative. Therefore, $\mathcal L(x_0,\eta_0)-\inf_{x,\eta} \mathcal L(x,\eta)$ can be bounded by the initial value $\mathcal L(x_0,\eta_0)$, which can be calculated easily from the initialization.
> >
> > As we discussed in the response to weakness 2, Theorem 2 focuses on the deterministic setting, where we have access to all the data to calculate the full gradient. Later, in Section 3.3, we extend our studies to the stochastic setting. For example, in Theorem 3, $\beta_t=\min(\frac{1}{2L_0}, \frac{\epsilon}{L_0\|v_t\|})$ only depends on the stochastic gradient, and not on the full gradient anymore.
> >
> > **Weaknesses 4:** Thanks for your suggestion. We have changed our expressions in the revision.
> >
> > **Question 1:**  Thanks for the suggestion. We change the title to ``Revisiting Large-Scale Non-convex Distributionally Robust Optimization: Partially Generalized Smoothness" to highlight the major technical development of partially generalized smoothness in this paper.
> >
> > **Question 2:**  Thanks for the question.
> >
> > We study the same problem with [1] under the same conditions. However, we prove a much simpler yet more precise characterization of the dual function: partial generalized smoothness and partially generalized affine noise. This allows for a more fundamental understanding of the dual problem structure for non-convex DRO than the one in [1], and further enables simpler algorithmic design and elegant theoretical proofs.
> >
> > We are the first to design the variance-reduced method for DRO problems, reducing the sample complexity from $\mathcal O(\epsilon^{-4})$ to $\mathcal O(\epsilon^{-3})$. [1] does not have results for variance-reduced method and complexity. Our analysis for this part is completely new compared to [1].
> >
> > [1] Jikai Jin, Bohang Zhang, Haiyang Wang, and Liwei Wang. Non-convex distributionally robust optimization: Non-asymptotic analysis. In Proceedings of Advances in Neural Information Processing Systems, volume 34, pp. 2771–2782, 2021.
> >
> > **Question 3:**  Good question! Yes, your understanding is correct.
> > First of all, as we discussed in response to weakness 1, such a Lipschitzness assumption is assumed in all non-convex DRO papers.  Second,  if the loss function $\ell$ is not Lipschitz but generalized smooth, e.g., the model is RNN or LSTM, results in [1] do not generalize and the dual function does not satisfy the generalized smoothness. However, using our approach, i.e., separately analyzing the smoothness of $x$ and $\eta$, we could bound the Lipschitz constant in $term(a)$ by a function of $x$ and $\eta$, and then try to prove the convergence. However, this takes non-trivial efforts and could be of independent interest.
> >
> > [1] Jikai Jin, Bohang Zhang, Haiyang Wang, and Liwei Wang. Non-convex distributionally robust optimization: Non-asymptotic analysis. In Proceedings of Advances in Neural Information Processing Systems, volume 34, pp. 2771–2782, 2021.
> >
> > **Question 4:**  Thanks for the question.
> >
> >  Consider a simpler problem of using SGD to solve standard $L$-smooth problem with mini-batch, then we only need a constant step size $\beta_t$. Here, the design of  $\beta_t=\mathcal O(\min(\frac{\epsilon}{L_0\|v_t\|},\frac{1}{2L_0}))$ is due to the additional terms introduced by the partially generalized smoothness. Since we alternatively update $x$ and $\eta$, when we update $x$, the current $\eta$ may be far away from its optimal point, making both the gradient on $\eta$ and the Lipschitz constant to $x$ potentially unbounded. Thus the clipping term in the stepsize is required to solve this potentially unbounded  Lipschitz constant problem.
> >
> > In our proof, we first show $\mathbb E[\beta_t\|v_t\|^2]\le \mathcal O(\epsilon^2)+\beta_t^4\|v_t\|^4$. By setting $\beta_t=\mathcal O(\min(\frac{\epsilon}{L_0\|v_t\|},\frac{1}{2L_0}))$, we can further get $\mathbb E[\beta_t\|v_t\|^2]\le\mathcal O(\epsilon^2)$.
> > Moreover, it follows that $\beta_t\|v_t\|^2=\epsilon^2\min(\frac{\|v_t\|^2}{2L_0\epsilon^2},\frac{\|v_t\|}{L_0\epsilon})\ge \frac{\epsilon\|v_t\|}{L_0}-\frac{\epsilon^2}{2L_0}$ since for any $x\ge 0$ we have that $\min(\frac{x^2}{2},x)\ge x-\frac{1}{2}$. We then can show $\mathbb E[\frac{\epsilon\|v_t\|}{L_0}]\le \mathcal O(\epsilon^2)$.

---

> > > ### Author Response · Authors · 2024-11-24
> > >
> > > Dear Reviewer oGQa,
> > >
> > > As the discussion period is approaching its end, we have yet to receive your feedback on our responses. We would greatly appreciate it if you could confirm whether we have addressed your concerns.
> > >
> > > We would be happy to clarify or provide additional information if needed. Additionally, we would be grateful if you could reconsider the score based on our clarifications.
> > >
> > > Thank you for your time and attention.
> > >
> > > Best regards,
> > >
> > > The authors

---

> > > > ### Comment · Reviewer_oGQa · 2024-11-25
> > > >
> > > > I thank the reviewers' detailed response. I slightly increase my score.

---

> > > > > ### Author Response · Authors · 2024-11-25
> > > > >
> > > > > Dear Reviewer oGQa,
> > > > >
> > > > > Thanks so much for your reply! We are happy that our responses address your concerns.
> > > > >
> > > > > Best Regards,
> > > > >
> > > > > Authors

---

### Official Review · Reviewer_4jhU · 2024-11-04

**Soundness:** 3
**Presentation:** 3
**Contribution:** 2
**Rating:** 6
**Confidence:** 4

**Summary:**

In this paper, the authors consider a distributionally robust optimization (DRO) problem with non-convex smooth losses. The previous work of Jin et al studied the same problem and showed that the dual problem satisfies a notion of generalized smoothness and bounded variance. They then used a normalized momentum algorithm that provably converges to stationary points under these conditions. By contrast, this paper shows that one can provide a more fine-grained analysis of the smoothness and variance. The authors that propose alternative optimization methods based on these refined properties. Finally, experimental results show that the proposed methods outperform the algorithm in Jin et al.

**Strengths:**

1. The authors conduct an improved analysis of the non-convex DRO problem and the analysis leads to simpler algorithms and theory. The partial generalized smoothness condition might be of interest in other settings as well.

2. Theoretical results are presented in a clear manner, and the authors provide detailed comparison with the previous work by Jin et al, thereby hightlighting the contribution of this work.

3. Besides improving over the normalized momentum method in Jin et al, the authors also provide an extension of the well-known spyder algorithm to the DRO setting. I'm not aware of such type of existing results in the literature.

**Weaknesses:**

1. In some paragraphs the authors seem to misuse \citet instead of \citep (e.g. the first paragraph in page 4)

2. The authors only run experiments at a relatively small scale. It would be interesting to compare different methods by running a large-scale experiments.

**Questions:**

I do not have questions for this work.

---

> ### Author Response · Authors · 2024-11-21
>
> We thank the reviewer 4jhU for the valuable feedback and here are our responses.
>
> **Weakness 1:**  Thanks for pointing it out. We have corrected this error in the revision.
>
> **Weakness 2:**  Thanks for the question. We have provided a large-scale experiment in Section A.10.2 in the revision. We can observe that the Normalized-SPIDER and our D-SPIDER-C have similar performance and converge faster than other methods.
> Our D-SGD-C has a similar performance compared with the Normalized-SGD with momentum method but our D-SPIDER-C outperforms the Normalized-SGD with momentum method.

---

> > ### Author Response · Authors · 2024-11-24
> >
> > Dear Reviewer 4jhU,
> >
> > As the discussion period is approaching its end, we have yet to receive your feedback on our responses. We would greatly appreciate it if you could confirm whether we have addressed your concerns.
> >
> > We would be happy to clarify or provide additional information if needed. Additionally, we would be grateful if you could reconsider the score based on our clarifications.
> >
> > Thank you for your time and attention.
> >
> > Best regards,
> > The authors

---

> > > ### Comment · Reviewer_4jhU · 2024-11-24
> > >
> > > Thank you for your reply. I don't have any further questions for you and I'll maintain my score.

---

> > > > ### Author Response · Authors · 2024-11-24
> > > >
> > > > Dear Reviewer 4jhU,
> > > >
> > > > Thanks so much for your reply! We are happy that our responses address your concerns.
> > > >
> > > > Best Regards, Authors

---

### Meta-Review · Area_Chair_c7pM · 2024-12-20

**Metareview:**

The paper studies solving f-divergence penalized nonconvex DRO problems, where the loss function is assumed to be both Lipschitz and smooth. The paper relies on the dual formulation of such DRO problems, which turns the original DRO problem -- which is a min-max optimization problem -- into a minimization only problem over two sets of variables. The proposed algorithms apply to this dual formulation, min-only problem. The algorithms themselves are all based on alternating gradient descent updates (possibly with added variance reduction) over the two sets of variables. On the analysis side, the paper proves a generalized smoothness condition and a structural result about the variance for the dual problem, which they rely upon to obtain the claimed complexity results. Without variance reduction, the oracle complexity is $1/\epsilon^4$, which matches prior results, whereas with (SPIDER) variance reduction the result improves to $1/\epsilon^3$.

While there were initial concerns about the paper and limited enthusiasm about the results, the reviewers eventually all leaned towards acceptance.

**Additional Comments On Reviewer Discussion:**

The review khoB was downweighted due to being brief and uninformative.

The discussion with the remaining reviewers centered around a few main points: (1) large batch sizes, (2) experimental details, (3) assumptions, (4) novelty/comparison to prior work. For (1), the authors argued that the use of large batch sizes in their work is mild, since prior work required computing the full gradient at initialization. For (2), I am less concerned about experiments for theoretical papers, though it seems like the authors did manage to convince the reviewer there were sufficient experimental setups. For (3), even though the reviewer oGQa seemed convinced and increased their score, I am less convinced by the authors' response. First, for the step size requirement of being inversely proportional to the function value gap, the authors argue this is mild because they can bound the gap *above.* But actually they would need to bound the gap *below,* since they need to bound the step size *above.* I did not weigh this much because there are other results that do not need this assumption, but I hope the authors will address it appropriately. For the strong assumptions about the loss function, I found the authors' response somewhat convoluted and misleading. In particular, they made a claim that all prior work needed the same assumptions, but this is not entirely true. They quoted results for convex optimization, which were indeed requiring the loss to be Lipschitz but they did not require it to be smooth at the same time. Additionally, the paper [5] in the response did not require Lipschitzness. For (4), the authors argued about novelty via their generalized Lipschitz condition.

---

### Decision · Program_Chairs · 2025-01-22

Accept (Poster)